# Mechanodetection of neighbor plants elicits adaptive leaf movements through calcium dynamics

Chrysoula K. Pantazopoulou [1] ✉, Sara Buti [1], Chi Tam Nguyen [2], Lisa Oskam [1], Daan A. Weits [1], Edward E. Farmer [2], Kaisa Kajala [1] & Ronald Pierik [1,3] ✉

Plants detect their neighbors via various cues, including reflected light and touching of leaf tips, which elicit upward leaf movement (hyponasty). It is currently unknown how touch is sensed and how the signal is transferred from the leaf tip to the petiole base that drives hyponasty. Here, we show that touch-induced hyponasty involves a signal transduction pathway that is distinct from light-mediated hyponasty. We found that mechanostimulation of the leaf tip upon touching causes cytosolic calcium ($[Ca^{2+}]_{cyt}$ induction in leaf tip trichomes that spreads towards the petiole. Both perturbation of the calcium response and the absence of trichomes reduce touch-induced hyponasty. Finally, using plant competition assays, we show that touch-induced hyponasty is adaptive in dense stands of Arabidopsis. We thus establish a novel, adaptive mechanism regulating hyponastic leaf movement in response to mechanostimulation by neighbors in dense vegetation.

Plants growing at high densities compete for resources, including light. Light quality changes are generally known to be received as a neighbor detection cue. The best-established above-ground neighbor detection signal is the reduction of the red (R) to far-red (FR) light ratio (R:FR)[1] following from waveband-specific absorption (R) and reflection (FR) properties of leaves. Plants respond to reduced R:FR through shade avoidance syndrome (SAS) responses[2,3], with a central role for PHYTOCHROME INTERACTING FACTORS (PIFs)[4–6]. SAS includes petiole, stem, and hypocotyl elongation, apical dominance, early flowering, and hyponasty (upward movement of the leaf)[7–9]. Interestingly, different responses are elicited, depending on where low R:FR is perceived: Local-FR enrichment of the leaf tip induces differential petiole elongation in the abaxial side of the petiole causing hyponasty, while local FR enrichment of the petiole leads to petiole elongation without hyponasty[10]. Although FR-enrichment by neighbors is a ubiquitously occurring early neighbor detection cue, it is complemented, and sometimes even preceded by, another neighbor detection cue in rosette plant canopies: touching of neighboring leaf tips[11].

In response to this mechanostimulation of the leaf tip a differential growth response is triggered in the petiole leading to hyponasty, reminiscent of the low R:FR-induced hyponastic response. This hyponastic leaf movement creates a vertical canopy structure that then generates the classic FR light reflection leading to R:FR signaling in plants[11]. The mechanisms involved in detecting and spatially relaying the mechanostimulation from leaf tip to base are currently unknown. Mechanostimulation responses and wounding are primarily regulated by the plant hormone jasmonic acid (JA)[12,13]. Wound responses involve long-distance signaling between the wound and distal tissues that both elicit a rise in JA, and whose JA response patterns are connected via membrane depolarizations processed via Glutamate-receptor-like proteins (GLRs)[12,14–17]. Wounding, or mechanostimulation stimulates increases in cytosolic calcium $[Ca^{2+}]_{cyt}$ with the GLRs controlling this induction[13,17–22].

Here, we investigate how leaf tip touching is sensed and signaled over the leaf in order to induce differential growth in the petiole base. We show that the signaling mechanisms of touch-induced hyponasty

[1]Plant-Environment Signaling, Institute of Environment Biology, Utrecht University, Utrecht, The Netherlands. [2]Department of Plant Molecular Biology, University of Lausanne, Lausanne, Switzerland. [3]Laboratory of Molecular Biology, Wageningen University, Wageningen, The Netherlands. ✉e-mail: c.pantazopoulou@uu.nl; ronald.pierik@wur.nl

are fundamentally different from those involved in low R:FR light-mediated hyponasty. In a transcriptome survey, we observed strong enrichment of JA- and abscisic acid (ABA)-associated genes, whereas the canonical auxin pathway was not induced, unlike R:FR-mediated leaf movement. We associated the transcriptome signatures with mechanical stimulation responses, happening specifically in the leaf tip. Using the GFP-based GCaMP3 biosensor, we observed that mechanostimulation of the leaf tip promotes $[Ca^{2+}]_{cyt}$ induction and spread towards the petiole in a GLRs-dependent manner. Interestingly, this $[Ca^{2+}]_{cyt}$ increase is triggered from the trichomes, the very first tissue to interact between two touching leaves. We show that these are not just the first cells to contact neighbors; they also play an important role in inducing the response, since trichome-less mutants have strongly reduced touch-induced hyponasty.

## Results and discussion

### Distinct signaling pathways regulate touch- and FR-induced hyponasty

In the non-vertically structured canopy of a young Arabidopsis stand, touching neighboring leaves is the earliest mode of above-ground neighbor detection, eliciting hyponasty[11]. This response entails approximately 20 degrees of upward movement in 24 h exposure against an inert transparent tag mimicking a neighbor leaf (Fig. 1a, b and Supplementary Fig. 1a)[11], and this is further increased after 48 h (Supplementary Fig. 1a). Similar responses are observed in the unrelated species *Nicotiana benthamiana* where the response also remains local to the leaf perceiving touch, (Supplementary Fig. 1b–d), indicating that touch-induced hyponasty is not restricted to *Arabidopsis*. We have recently shown that local reduction of phytochrome activity in the leaf tip through local FR enrichment, on approximately the same position as where leaf-leaf mechanical interactions occur, induces a similar magnitude of hyponasty[10] (Fig. 1c). This FR-induced hyponasty acts through PHYTOCHROME INTERACTING FACTOR (PIF)4, PIF5 and PIF7 that activate the auxin pathway, at least partially through *YUCCA* (*YUC)8* and *YUC9* gene expression[10]. The resulting elevated auxin is transported and indeed *pin3pin4pin7* triple mutants are not hyponastic in response to FR treatment[10,23]. We, therefore, started out by verifying if this pathway is also activated to regulate touch-induced hyponasty. Interestingly, the severe shade avoidance mutants *pif4pif5*, *pif7*, *pin3pin4pin7*, and *yuc2yuc5yuc8yuc9*, all showed a wild-type hyponastic response to touch (Fig. 1d–f), whilst being fully irresponsive to FR application (Fig. 1g–i). Another auxin biosynthesis mutant *wei8/sav3* and the *yuc8* single mutant also showed a wild-type-like touch-induced hyponastic response (Supplementary Fig. 1e, f). Therefore, despite the phenotypic similarity of these two responses, the signaling pathway of touch-induced hyponasty is unique from the core shade avoidance pathway.

### Transcriptome analysis reveals specific regulation in petiole versus leaf tip

To unveil the mechanisms of touch-induced hyponasty in depth, we analyzed transcriptome data (using Affymetrix *Arabidopsis* Gene 1.1 ST arrays), comparing the site of perception (leaf tip) and the site of action (petiole base) (Fig. 2a) under control and touch conditions. We found similar numbers of up- and downregulated genes between the two tissues (Fig. 2b), but interestingly there was nearly no overlap in differentially expressed genes (DEGs) (Fig. 2c), indicating that different parts of the leaf have distinct transcriptional responses to touch. We then compared the touch-induced hyponasty transcriptome data with the previously published local FR-induced hyponasty transcriptome data that were collected from the same leaf tissues and under identical conditions in the same run of experiments[10]. Touch induced only a minor number of DEGs (less than 100) compared to the low R:FR treatment (over 700 DEGs) in the leaf tip, and also in the petiole base the number of DEGs in low R:FR treatment is almost 5 times higher than

in the touch treatment (Supplementary Fig. 2a). Amongst the leaf tip DEGs only 35 genes overlapped between touch and low R:FR treatments. Although this is approximately half of the touch-induced genes, the genes themselves (Supplementary Data 1) are not typically associated with the canonical shade avoidance machinery, but rather with abscisic acid (ABA) response, jasmonic acid (JA) metabolism and cell wall homeostasis (Fig. 2d, Supplementary Fig. 2b). Next, we compared the touch-induced DEGs from the tip with previously published mechanostimulation-induced DEGs[24–26] (Supplementary Fig. 3, Supplementary Data 2). We found significant overlap with brushing and *GL1*-dependent and independent response[24,26] (Supplementary Fig. 3a, b) while there was no significant overlap with *MYC2 MYC3 MYC4*-dependent water spray response and the MYC2-regulon[25] (Supplementary Fig. 3c, d, Supplementary Data 2).

Since the Gene ontology (GO) enrichment analysis indicated enrichment of JA and ABA-related processes in the leaf tip and petiole base (Fig. 2d), we tested the role of JA on touch-induced leaf hyponasty. Application of exogenous MeJA (50 and 100 μM Methyl jasmonate) to the leaf blade did not inhibit touch-induced hyponasty. However, higher concentration (200 μM) of MeJA partially suppressed hyponasty maybe due to the decrease in petiole elongation (Supplementary Fig. 4a). The touch response was not different from the wild-type in the JA biosynthesis mutants *lox2*, *loxQ* (*lox2lox3lox4lox6*) and *aos* and in the JA receptor mutant *coi1-34* (Supplementary Fig. 4b, c). MYC transcription factors act downstream of JA activation and although the *myc* single mutants were similar to the wild-type, the *myc2myc3myc4* triple mutant had a mildly reduced response (Supplementary Fig. 4d, e) with a similar local-FR-induced hyponasty response (Supplementary Fig. 4f). Since the transcriptome analysis also revealed ABA signatures in the petiole base, we first applied ABA to the petiole, which reduced hyponasty only at the highest concentration (Supplementary Fig. 4g), probably due to an overall inhibition of petiole growth (Supplementary Fig. 4h). Mutants for ABA biosynthesis (*aba2-1*, *aba3-1*), ABA perception (*pyr1pyl1pyl2pyl4*, referred to as *abaQ*) and ABA signaling (*areb1areb2abf3abf1*, referred to as *arebQ*) all responded similarly to Col-0 wild-type (Supplementary Fig. 4i–k). Summarizing, ABA and MeJA application can limit petiole growth and thereby hyponasty, but a wide variety of mutants for the JA and ABA pathways suggest no major role for these two hormones in touch-induced hyponasty.

### Touch-induced hyponasty is mediated via changes in cytosolic $[Ca^{2+}]$

Mechanostimulation responses are highly associated with cytosolic calcium dynamics ($[Ca^{2+}]_{cyt}$) following mechanical perturbation in many species, including *Arabidopsis*[17,21,27–29]. Specifically, mechanical perturbation has been found to increase intracellular calcium by triggering temporary changes in the cytosolic calcium concentration[18,22,30,31]. Indeed, our transcriptome data showed GO enrichment of calcium ion transport in the leaf tip upon touch (Fig. 2d). To verify if $[Ca^{2+}]_{cyt}$ is affected during touch-induced hyponasty, we used the GFP fluorescence-based cytosolic calcium biosensor *UBQ10pro::GCaMP3* that allows detection of $[Ca^{2+}]_{cyt}$ fluxes in the leaf[17,28]. Upon gently touching the 5th youngest leaf of 4-week-old *Arabidopsis* we recorded the GCaMP3 fluorescence dynamics. We measured GCaMP3 fluorescence with a microscope positioned above the leaf, in the leaf tip (position 1), in two subsequent positions of the primary vein of the leaf (positions 2 and 3), the leaf blade-petiole junction (position 4) and the middle and base of the petiole (positions 5 and 6, respectively) (Fig. 3a, b). We observed that GCaMP3 fluorescence started to increase in 4 min (250 sec) upon gentle touch (Supplementary Movie 1) in all the positions except position 6 (Fig. 3c; Supplementary Table 1). This is substantially slower than what has been reported in wound or other touch responses, where changes were recorded as early as 30 seconds after stimulation[17,21,29,32]. Clearly, the mechanical force of stimulation in the

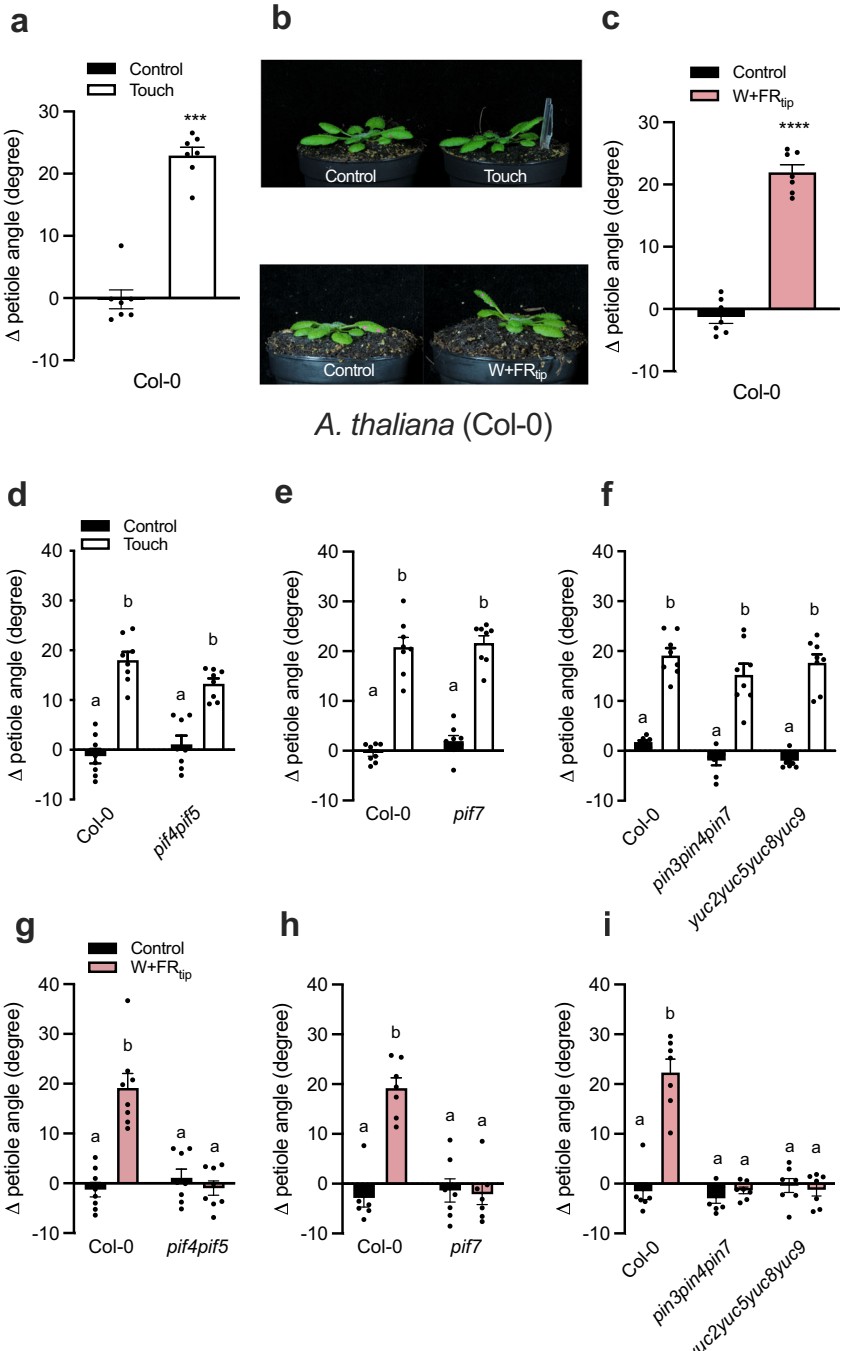

**Fig. 1 | Touch and Local-FR-induced hyponasty have a similar phenotypic response but different genetic basis.** Differential petiole angle of Col-0 after 24 h (**a**, **b**) of touch and (**b**, **c**) local-FR treatment. $n = 7$ independent biological replicates for **a**, **c**. Differential petiole angle of Col-0 compared to (**d**, **g**) *pif4pif5*, (**e**, **h**) *pif7*, (**f**, **i**) *pin3pin4pin7* and *yuc2yuc5yuc8yuc9* mutants after 24 h of **d**–**f** touch and **g**–**i** local FR treatment. $n = 8$ biologically independent replicates for **d**–**g** and $n = 7$ biologically independent replicates **h** and **i**. The "W+FR$_{tip}$" refers to the local FR treatment of the leaf tip and "Touch" refers to the touch treatment of the leaf tip with a gently positioned transparent tag next to the leaf. Data represent mean ± SE. Black dots represent the individual data. Different letters or asterisks indicate significant differences (two-way ANOVA with Tukey's post hoc test or unpaired $t$ test; $p < 0.05$).

experiments here is much lower than during severe wounding or even the rapid leaf closure of carnivorous Venus flytrap upon capture of the prey[33,34]. Possibly the degree or force of the mechanical stress could affect the rapidity of induction of $[Ca^{2+}]_{cyt}$ flux. Touching of the leaf tip (position 1) led to a first calcium wave at around 7 min (420 sec) with a peak GCaMP3 signal at 13 min (780 sec) upon touch in the leaf tip itself (position 1), followed by a gradual decrease of the fluorescence signal. These calcium waves progressed from the leaf tip through the leaf and into the petiole, but fluorescence intensity decreased with distance

from the site of mechanostimulation (Fig. 3c; Supplementary Table 1). The GCaMP3 fluorescence in the petiole (positions 5 and 6) was rather weak and this may indicate only very weak $[Ca^{2+}]_{cyt}$ signal reaching the base, or a preferential localization to the abaxial side that cannot be imaged in our microscope configuration. To verify that the observed $[Ca^{2+}]_{cyt}$ dynamics were required for touch-induced hyponasty, we used chemical inhibitors: the $Ca^{2+}$ channels inhibitor $LaCl_3$[21,29,35] and the $Ca^{2+}$ channel antagonist Verapamil[35,36] and monitored the $[Ca^{2+}]_{cyt}$ dynamics through time. Exogenously applied $LaCl_3$ and Verapamil at the leaf tip

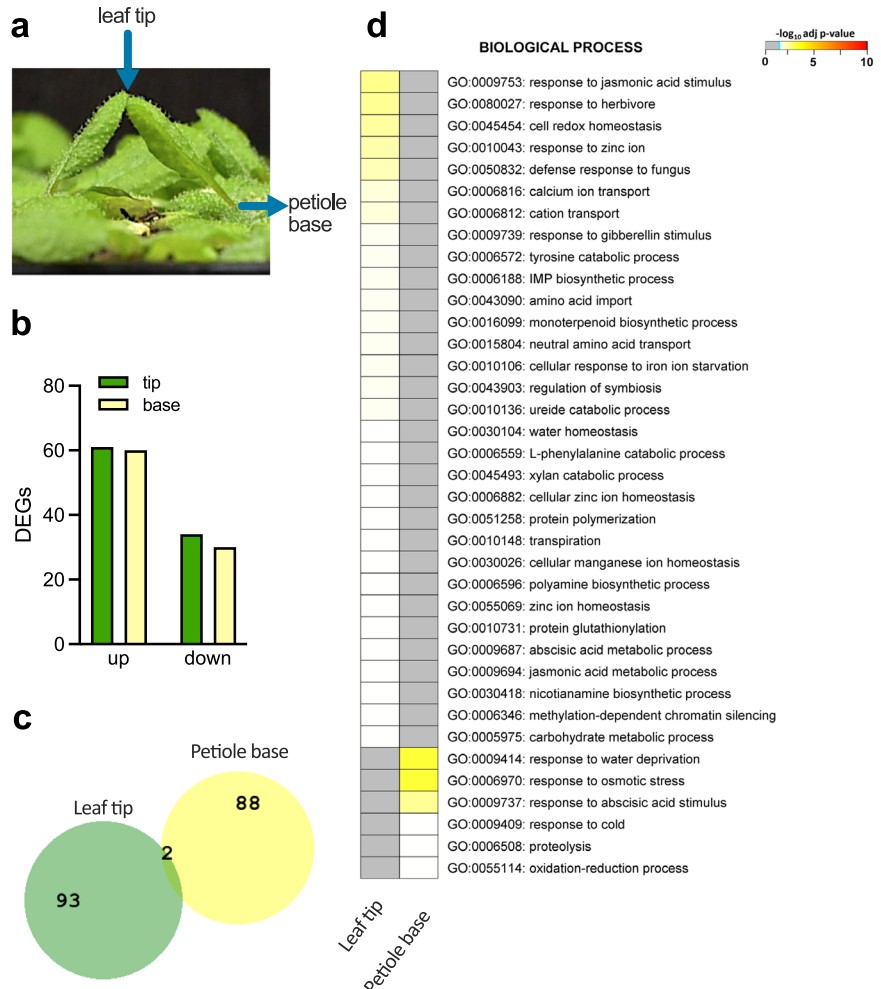

**Fig. 2 | Comparative analysis of touch-induced hyponasty in the leaf tip and the petiole base. a** Leaf tip and petiole base tissues were harvested, in the experiment touch was induced by an inert transparent tag. **b** Number of differentially expressed genes (DEGs) in the lamina tip and the petiole base. **c** Comparison of DEGs in leaf tip ("Leaf tip") and petiole base ("Petiole base") in response to touch (adj. *P* value < 0.05). **d** GO enrichment analysis of the DEGs in the leaf tip ("Tip") and the petiole base ("Base").

resulted in a strong reduction of the touch-induced $[Ca^{2+}]_{cyt}$ increase at positions 1–6, particularly at the higher concentration (Supplementary Fig. 5a–n). Physiological experiments with exogenous application of $LaCl_3$ and Verapamil on the touched leaf showed that these calcium inhibitors strongly reduced the touch-induced hyponastic response (Supplementary Fig. 5o, p). Although $LaCl_3$ and Verapamil treatment could give pleiotropic effects, we observed that $LaCl_3$ and Verapamil did not affect the hyponastic response to local FR treatment at all (Supplementary Fig. 5q, r), suggesting their effects to be specific to touch-induced hyponasty. In addition, we aimed to investigate whether an increase of $[Ca^{2+}]_{cyt}$ levels without a physical touch treatment could initiate a hyponastic response. For this purpose, we used the $Ca^{2+}$ agonist Mastoparan which is known to elevate $[Ca^{2+}]_{cyt}$ levels[37,38] (Supplementary Fig. 6a). Our results showed that exogenous application of Mastoparan induced a clear upward leaf movement (Supplementary Fig. 6b), indicating that $[Ca^{2+}]_{cyt}$ itself can act to initiate touch-induced hyponasty.

Clade 3 Glutamate-receptor-like proteins (GLRs) have been associated with the propagation of long-distance electrical signals via $[Ca^{2+}]_{cyt}$ after mechanical stress[12,17,21,22]. Nguyen et al.[17] showed that GLR3.1, GLR3.3, GLR3.6 are localized in the vasculature and a calcium signal is severely attenuated in the *glr3.1glr3.3* and *glr3.3glr3.6* double mutants. Since the calcium waves observed here (Fig. 3c) start from the leaf tip and move to the petiole along the primary vein, we investigated the involvement of these GLRs in touch-induced hyponasty. Indeed, the *glr3.3glr3.6* double mutant showed a slightly reduced touch-induced hyponasty (Fig. 4a). Given the rather weak effect, we also created a triple knockout mutant *glr3.1glr3.3glr3.6* and found that its touch-induced hyponasty was strongly attenuated (Fig. 4b, c). Furthermore, we monitored the $[Ca^{2+}]_{cyt}$ dynamics in the *glr3.1glr3.3glr3.6* by crossing *glr3.1glr3.3glr3.6* with *UBQ10pro::GCaMP3*. Imaging of the *glr3.1glr3.3glr3.6* with *UBQ10pro::GCaMP3* showed a decrease of the $[Ca^{2+}]_{cyt}$ signal in all the monitored positions of the touched leaf (Supplementary Movie 2; position 1–6; Fig. 4d, e and Supplementary Fig. 7). Collectively these data indicate that clade 3 GLRs may redundantly regulate touch-induced hyponasty. This triple mutant has wild-type-like hyponasty in response to local FR treatment (Supplementary Fig. 6c), reinforcing the specificity of the $[Ca^{2+}]_{cyt}$-associated regulatory route for mechanostimulation.

## Trichomes are involved in touch-induced hyponasty

Trichomes on the leaf edge are the first cells to physically interact with neighboring leaves when they grow towards each other in dense stands. Previous and recent work established that trichomes can trigger calcium oscillations in the trichome stalk and the trichome base cells surrounding the trichomes upon mechanical perturbation[26,39]. We monitored GCaMP3 fluorescence at three trichome positions upon gentle trichome touching: the trichome stalk (position 1, Fig. 5a, b), the

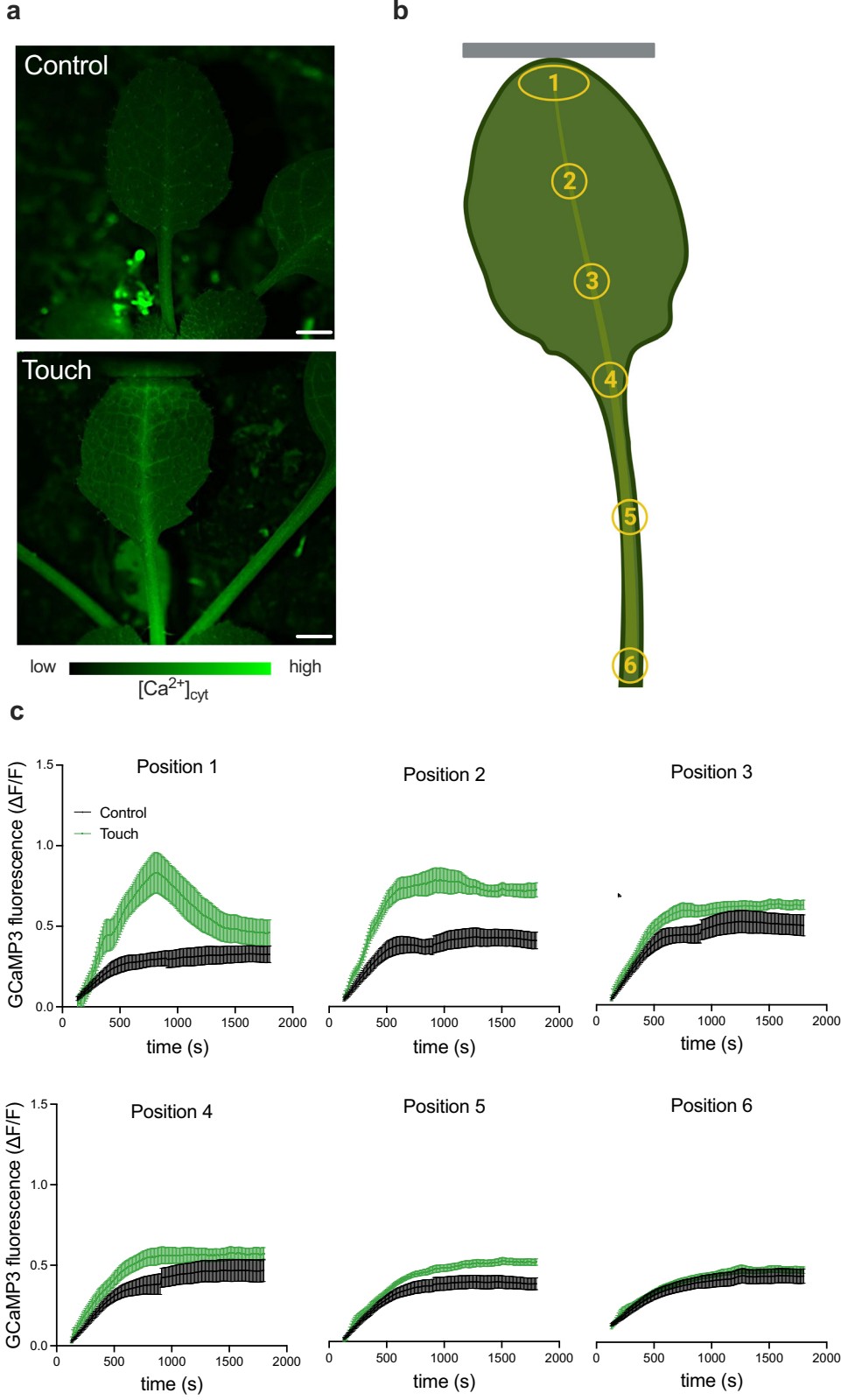

**Fig. 3 | Touch-induced hyponasty causes cytosolic Ca²⁺ increase in the touched leaf. a** Fluorescence induction in the leaf of untouched (Control) or touch-treated (Touch) leaves after 10 min of touch using the fluorescent cytosolic calcium bio-sensor *UBQ10pro::GCaMP3*. **b** Six different positions were used to measure the GCaMP3 fluorescence in leaf upon control or touch treatment. Gray line represents the transparent tag. **c** Time course of GCaMP3 fluorescence intensity in leaf tip (position 1), primary vein of the lamina (position 2 and 3), lamina-petiole junction (position 4), in the middle of the adaxial site of the petiole (position 5), and in the adaxial site of the petiole base (position 6) upon control (black line) and touch (green line) treatment. For "Control" treatment $n = 10$ biologically independent replicates while for "Touch" treatment $n = 11$ biologically independent replicates. Touch treatment started at $t = 2$ min (120 sec) and fluorescence was followed for 25 minutes. Touch was induced by a gently positioned transparent tag next to the leaf. Data represent mean ± SE. Scale bar corresponds to 1 mm. Leaf drawing in **b** was created with Biorender.

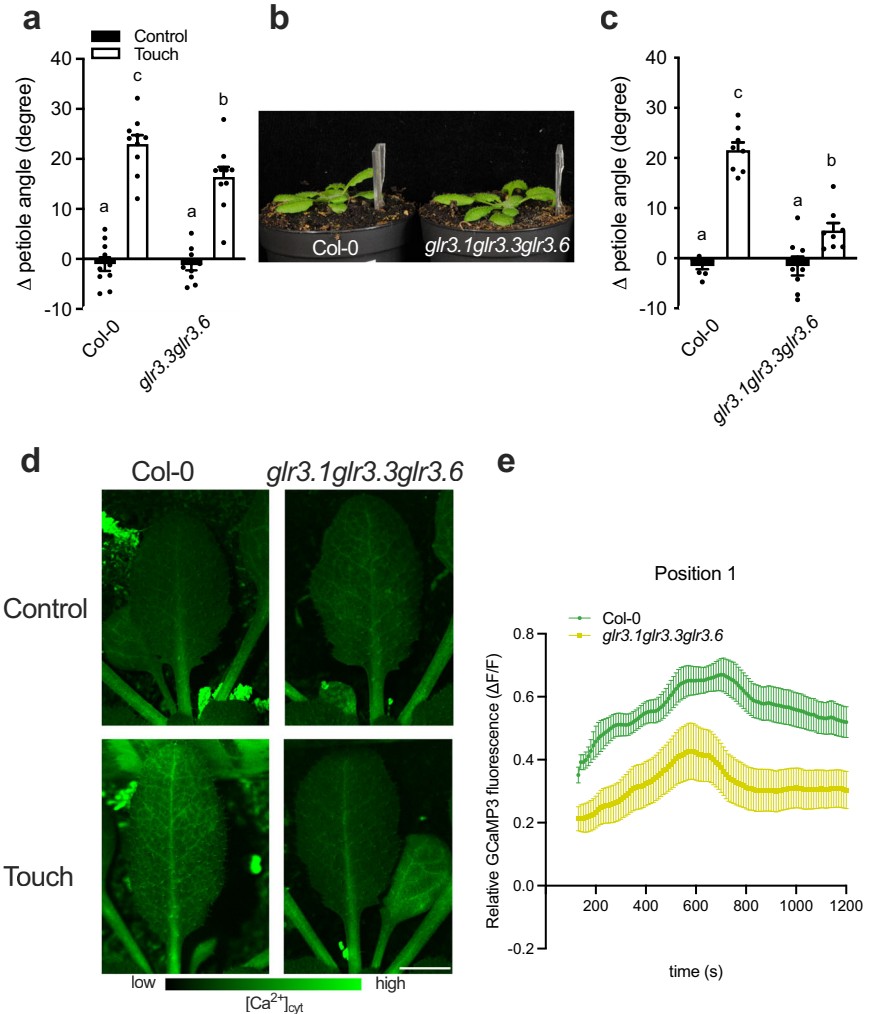

**Fig. 4 | GLRs are required for the full touch-induced hyponasty response with the involvement of $[Ca^{2+}]_{cyt}$. a** Differential petiole angle of Col-0 compared to *glr3.3glr3.6* double mutant after 24 h of touch treatment. $n = 10$ biologically independent replicates (**b, c**). The touch response of the *glr3.2glr3.3glr3.6* triple mutant is significantly reduced compared to Col-0 after 24 h of touch treatment. $n = 8$ biologically independent replicates. **d** Picture of fluorescence induction in the leaf after 12 min of untouched (Control) or touch treatment (Touch) using the fluorescent cytosolic calcium biosensor *UBQ10pro::GCaMP3* and *glr3.1glr3.3glr3.6* with *UBQ10pro::GCaMP3*. **e** Relative GCaMP3 fluorescence (ΔF/F touch treatment$_{tx}$−ΔF/F control$_{averagetx}$) time course in leaf tip (Position 1) touch treatment in Col-0 and *glr3.1glr3.3glr3.6*. **d, e** $n = 12$ biologically independent replicates for *UBQ10-pro::GCaMP3* and $n = 15$ biologically independent replicates for *glr3.1glr3.3glr3.6* with *UBQ10pro::GCaMP3*. Touch treatment started at $t = 2$ min (120 sec) and fluorescence was followed for 20 minutes. Touch was induced by a gently positioned transparent tag next to the leaf. Black dots represent the individual data. Data represent mean ± SE. Different letters indicate significant differences (two-way ANOVA with Tukey's post hoc test; $p < 0.05$). Scale bar corresponds to 4 mm.

trichome base cells (position 2), and the epidermal cells around the trichome (position 3) (Supplementary Fig. 8a–c). We also monitored how far the calcium can spread upon trichome-touch by recording an area adjacent to the neighboring trichome (position 4, Supplementary Fig. 8a, d). A strong induction of the GCaMP3 fluorescence was detected in the trichome stalk (position 1) 30 sec after very gently touching the trichome with a toothpick (Supplementary Movie 3), which rapidly spread to the base cells (position 2) and the epidermal cells around the trichome (position 3) (Fig. 5a, b, Supplementary Fig 8b, c). Interestingly, GCaMP3 fluorescence was detected near the neighboring trichome (position 4) suggesting that the $[Ca^{2+}]_{cyt}$ induction was strong enough to extend laterally to the untouched trichome area (Supplementary Fig. 8d). We then touched trichomes near the main vein and measured the GCaMP3 fluorescence changes in trichomes itself and the main vein part near that trichome (Fig. 5c, d and Supplementary Fig. 8e). Our measurements revealed increased [Ca2+]$_{cyt}$ in both the trichomes and the main vein. The [Ca2+]$_{cyt}$ fluorescence remained elevated in the main vein even after 120 sec

(Supplementary Fig. 8e). Notably, when we touched a trichome of *glr3.1glr3.3glr3.6* with *UBQ10pro::GCaMP3*, the rapid increase of $[Ca^{2+}]_{cyt}$ within the first 10 sec, disappeared very rapidly, leaving a much-reduced response as compared to wild-type Col-0 (Fig. 5e, f, and Supplementary Fig. 8f). Our findings indicate that gentle touching of just the trichomes can lead to calcium induction and spreading across the leaf blade, consistent with our observations that leaf tip touch can elicit a $[Ca^{2+}]_{cyt}$ wave along the leaf blade towards the petiole that could probably be GLR3.1, GLR3.3 and GLR3.6-dependent.

Since trichomes are the first cells to touch neighboring leaves, and they can generate $[Ca^{2+}]_{cyt}$ changes that progress through the leaf, we monitored the $[Ca^{2+}]_{cyt}$ changes in the *gl1* mutant that lacks trichomes in a genetic cross of *gl1* and *UBQ10pro::GCaMP3*. We found a compromised touch-induced $[Ca^{2+}]_{cyt}$ signal in all six leaf positions (from the leaf tip to the petiole base; Supplementary Movie 4; Fig. 6a, b and Supplementary Fig. 9a–e) as compared to Col-0. If trichomes-derived $[Ca^{2+}]_{cyt}$ would contribute to the hyponastic response, repeated brushing of just the leaf tip trichomes during the photoperiod might

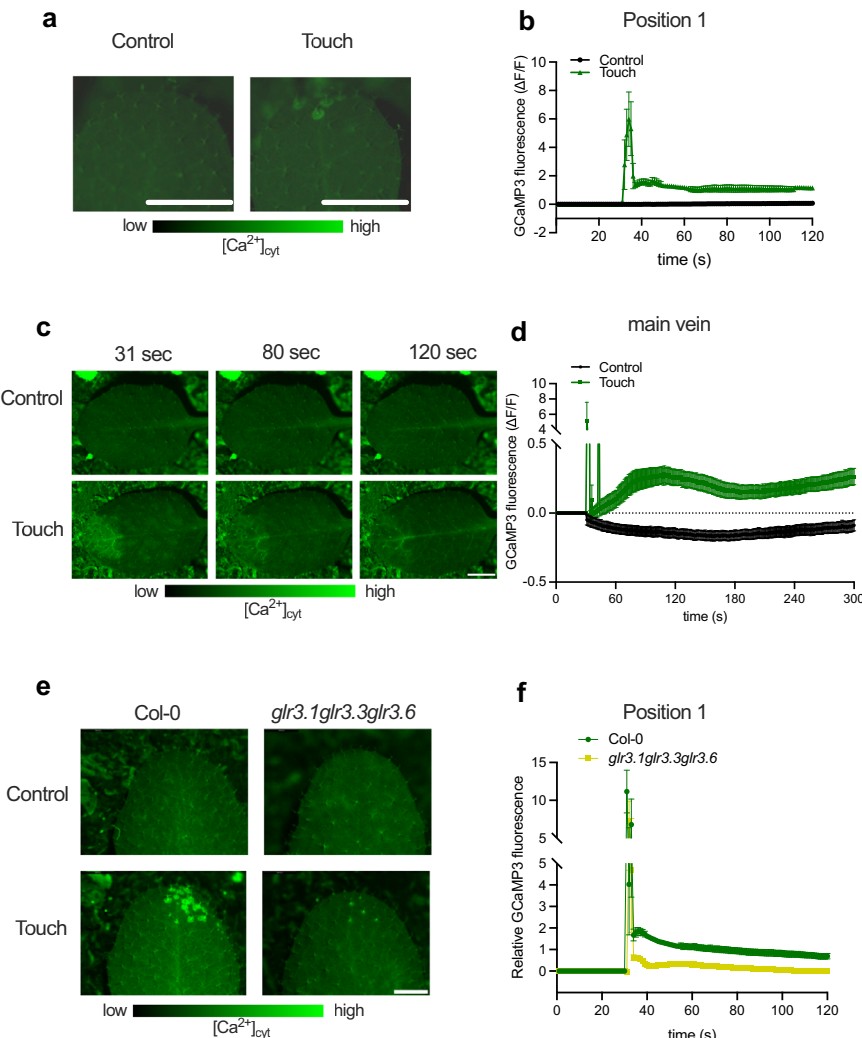

**Fig. 5 | Mechanostimulation in trichomes induces cytosolic calcium [Ca²⁺]$_{cyt}$ via a GLR-dependent pathway. a** GCaMP3 fluorescence in trichomes of untouched (Control) or touch-treated plants (Touch), using the fluorescent cytosolic calcium biosensor *UBQ10pro::GCaMP3*. **b** Time course of GCaMP3 fluorescence in trichome stalk of Col-0 upon control or touch treatment. For "Control" treatment, $n = 12$ biologically independent replicates, and for "Touch" treatment, $n = 10$ biologically independent replicates. Touch treatment started at $t = 30$ sec and fluorescence was followed for 2 minutes. To induce Touch, trichomes were gently touched with a toothpick. **c** GCaMP3 fluorescence in the main vein after 31, 80, and 120 seconds of untouched (Control) or touch treatment (Touch) of trichomes near the main vein, using the fluorescent cytosolic calcium biosensor *UBQ10pro::GCaMP3*. **d** Time course of GCaMP3 fluorescence in main vein of Col-0 upon control ($n = 18$) or touch ($n = 21$) treatment. For "Control" treatment, $n = 18$ biologically independent

replicates, and for "Touch" treatment, $n = 21$ biologically independent replicates. Touch treatment started at $t = 30$ sec and fluorescence was followed for 5 minutes. To induce Touch, trichomes were gently touched with a toothpick. **e** GCaMP3 fluorescence in trichomes after 10 seconds of untouched (Control) or touch treatment (Touch), using the fluorescent cytosolic calcium biosensor *UBQ10-pro::GCaMP3* and *glr3.1glr3.3glr3.6* with *UBQ10pro::GCaMP3*. **f** Relative GCaMP3 fluorescence (ΔF/F touch treatment$_{tx}$ − ΔF/F control$_{averagetx}$) time course in main vein after touch treatment in Col-0 and *glr3.1glr3.3glr3.6*. $n = 12$ biologically independent replicates for each genotype. Touch treatment started at $t = 30$ sec and fluorescence was followed for 2 minutes. To induce Touch, trichomes were gently touched with a toothpick. Data represent mean ± SE. Scale bars correspond to 1 mm (**a**), 3.5 mm (**c**), and 2 mm (**e**).

induce a (partial) hyponastic response, and a mild induction of hyponasty is indeed what we observed (Supplementary Fig. 9f). To further verify the role of trichomes in touch-induced hyponasty, we compared the Col-0 accession with the glabrous (non-trichome-forming) *Arabidopsis* accession 9354 (N28001)[40]. Accession 9354 indeed had a strongly reduced touch-induced hyponasty, as compared to Col-0 (Fig. 6c, d). Similar findings were also obtained using other glabrous *Arabidopsis* accessions (Fran-3, Wil-2, Br-0) upon touch treatment (Supplementary Fig. 10a). The *TTG1*[41] and *GL1*[42] genes are positive regulators of *Arabidopsis* trichomes, and their respective mutants, *ttg1* and *gl1*, do not form trichomes. These two mutants displayed severely reduced touch-induced hyponasty as well (Fig. 6e, f). At the same time, all the glabrous genotypes had wild-type-like local FR-induced hyponasty (Supplementary Fig. 10b–e). Although these data are a strong

indication that trichomes are indeed important for touch-induced leaf movement, glabrous mutants such as *gl1* and *ttg1* have also been linked to defects in cuticle formation[43]. We, therefore, also studied touch-induced hyponasty in two independent mutants that have established cuticle formation defects, but that have normal trichomes; *ecerefirum1* (*cer1*[26,43,44]) and *cer3*[26,43] in Col-0 and L*er* background, respectively. Both these mutants with defective cuticle formation displayed a WT-like touch-induced hyponastic response (Supplementary Fig. 10f, g), suggesting that aberrations in cuticle formation do not necessarily affect touch-induced hyponasty. We conclude that trichomes are specifically involved in touch-induced hyponasty.

After establishing the trichome - [Ca²⁺]$_{cyt}$ mechanism involved in touch-induced leaf movement, an important question remained: Is the touch-induced hyponasty of quantitative importance to plant

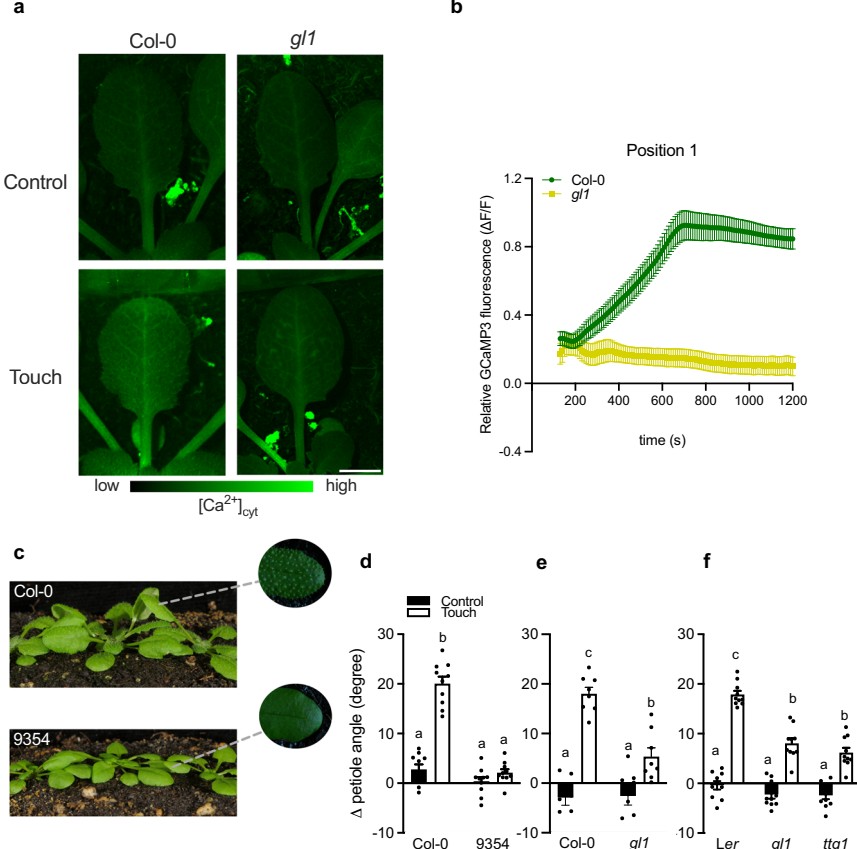

**Fig. 6 | Trichomes are required for touch-induced hyponasty. a** GCaMP3 fluorescence in trichomes after 10 min of touch treatment (Touch) or in untouched (Control) plants. Cytosolic $Ca^{2+}$ was monitored using the fluorescent biosensor *UBQ10pro::GCaMP3* in the wild-type and in *gl1*. Touch was induced by a gently positioned transparent tag next to the leaf. Relative GCaMP3 fluorescence ($\Delta$F/F touch treatment$_{tx}$−$\Delta$F/F control$_{averagetx}$) time course in leaf tip (Position 1) after touch treatment in Col-0 and *gl1*. For Col-0, $n = 10$ biologically independent replicates and for *gl1*, $n = 10$ biologically independent replicates. Touch treatment started at $t = 2$ min (120 sec) and fluorescence was followed for 20 minutes.

**c** Hyponastic responses are seen following leaf touching of densely grown Col-0 (with trichomes) but not in 9354 (without trichomes) plants. **d, e** Differential petiole angle of several genotypes that lack trichomes compared to their wild-type after 24 h of touch treatment: **d** Col-0 and 9354 and **e** Col-0 and *gl1*) and **f** L*er*, *ttg1* and *gl1*. $n = 10$ independent biological replicates for **d**, **f** while $n = 8$ independent biological replicates for **e**. Touch was induced by a gently positioned transparent tag next to the leaf. Black dots represent the individual data. Data represent mean ± SE. Different letters or asterisks indicate significant differences (two-way ANOVA with Tukey's post hoc test or *t* test; $p < 0.05$). Scale bar corresponds to 4 mm.

performance in dense stands? To answer this question we grew single plants, as well as monoculture stands and 1:1 mixed stands of the accessions Col-0 and 9354, Col-0 and Fran-3, Col-0 and Wil-2 as well as Col-0 and *gl1*, L*er* and *gl1* and L*er* and *ttg1* mutant. In the competition assays we kept root systems separated to prevent belowground competition, and measured shoot dry weight as a proxy of plant performance (Fig. 7a–d). Importantly, all genotypes and accessions showed growth similar to Col-0 when grown in crowded monocultures of each of the genotypes, but in the mixture where they interacted aboveground, Col-0 or L*er* outcompeted the glabrous accessions and mutants (Fig. 7, Supplementary Fig. 11 and Supplementary Fig. 12a). The superior competitive position of trichome-forming plants as compared to glabrous genotypes is consistent with results from independent experiments where we quantified the relative positions of interacting leaves in pairs of the same genotypes as in the competition assays. We observed that trichome-forming genotypes typically have a superior leaf position against glabrous genotypes in 1:1 interactions (see below). We observed similar results in canopies and 1:1 interactions of Col-0 and *glr3.1glr3.3glr3.6* (Supplementary Fig. 12b, c), indicating that not just trichomes, but also the transmission of a calcium signal is required for optimal plant performance in dense stands. We, therefore, propose that the trichome-dependent detection of neighboring leaves and the corresponding hyponasty, promote plant performance in competition for light. Suboptimal ability to do so, as in the 9354 accession, leads to

reduced competitive performance. Future studies may investigate the targets of trichome-derived $[Ca^{2+}]_{cyt}$ for differential growth between the abaxial and adaxial side of the petiole base, presumable involving regulators other than those involved in R:FR responses. Since hyponasty enhances the vertical element of a rosette canopy structure that then stimulates FR reflection to neighbors[10], the novel signaling route through trichomes and $[Ca^{2+}]_{cyt}$ dynamics precedes the better-known photoreceptor-driven pathways that are activated once a notable vertical canopy structure has been created.

## Methods
### Plant materials, growth, and measurements
Genotypes used in this study that are in the Col-0 background are: *pif4-101 pif5-1*[45], *pif7-1*[46], *pif4-101pif5-1pif7-1*[4], *abaQ*, *aba2-1*, *aba3-1*[47], *arebQ*[48], *myc2*[49], *myc3*, *myc4*, *myc2myc3myc4*[50], *glr3.3aglr3.6a*[12], *glr3.1aglr3.3aglr3.6a* were generated by crossing *glr3.3aglr3.6a* and *glr3.1aglr3.3a*[16], *aos*[51], *coi1-34*[52], *lox2*[15], *loxQ*[53], *wei8*[54], *pin3-3pin4pin7*[55], *yuc2yuc5yuc8yuc9*[56], *yuc8*[57], *gl1*[58], *cer1* (N660700), *cer3* (N520265) and *cer3* (N534318). Genotypes used in this study in the L*er* background are: *gl1* and *ttg1*[59], *cer1* (N31), *cer3* (N33). We also used *Arabidopsis* accessions without trichomes 9354 (N28001, point mutation[40]), Fran-3 (N75673, 1 bp insertion[40]), Wil-2 (N1596, locus deletion[40]), Br-0 (N22628, 1 bp deletion[40]) and *N. benthamiana*. Seeds were sown on Primasta® soil and stratified for 3 days (dark, 4 °C), before transferring

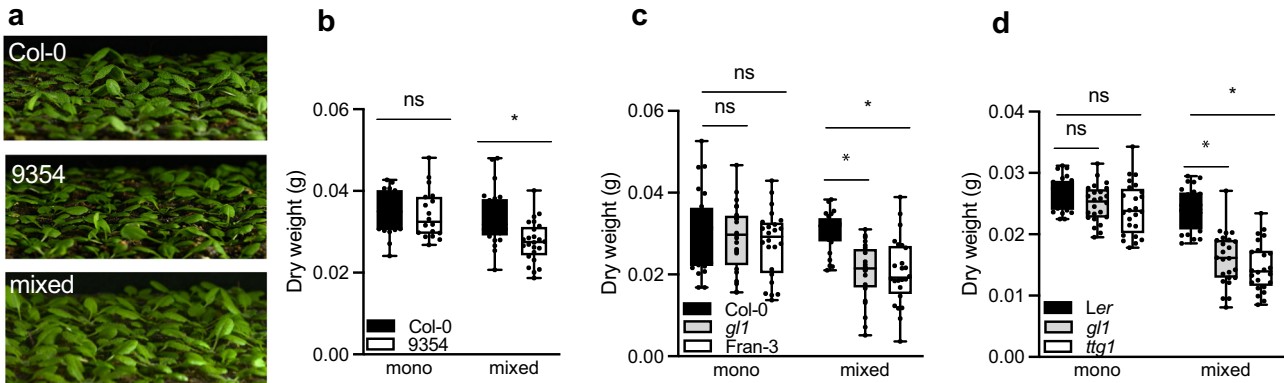

**Fig. 7 | Trichomes promote plant performance in competition for light.**
**a** Pictures illustrating (from the top to the bottom) the monoculture canopy of Col-0 and 9354 and the mixed canopy of both genotypes together. **b–d** Dry weight of **b** Col-0 and 9354, **c** Col-0 and *gl1*, Col-0 and Fran-3 **d** L*er* and *gl1*, L*er* and *ttg1* growing in monoculture canopies and in a mixed canopy (Col-0 together with 9354, Col-0 together with *gl1*, Col-0 together with Fran-3, L*er* together with *gl1* and L*er* together with *ttg1*). **b** n = 18 independent biological replicates for Col-0 and n = 24 independent biological replicates for 9355. **c**, **d** n = 24 independent biological replicates per genotype. The boxes show the inter-quartile ranges and the whiskers show minimum and maximum values but also plot the individual values as dots. The line in the middle of the box represents the median. Black dots represent the individual data. Data represent mean ± SE. Asterisks indicate significant differences (unpaired *t* test; *p* < 0.05).

to short day (9 h light / 15 h dark) growth rooms (130-135 μmol m⁻² s⁻¹ PAR, R:FR 2.3, 20 °C, 70% RH). For the $[Ca^{2+}]_{cyt}$ dynamics data we used the *UBQ10pro::GCaMP3 (Col-0), gl1* with *UBQ10pro::GCaMP3* (generated by crossing *UBQ10pro::GCaMP3 (Col-0) with gl1), glr3.1glr3.3glr3.6* with *UBQ10pro::GCaMP3* (generated by crossing *UBQ10pro::GCaMP3 (Col-0) with glr3.1glr3.3glr3)*. After 11 d, seedlings were transplanted to 70 ml pots for all experiments, except for the competition assays. For touch experiments an inert, transparent tag (polycyclic olefin) was placed in the soil, next to the fifth-youngest leaf of the 28-days old plant to mimic leaf-leaf touching (Supplementary Fig. 13a, b). Petiole angles to the horizontal of the fifth-youngest leaf that just touched the transparent tag were determined with image-J software (https://imagej.nih.gov/ij/download.html) from pictures taken just before (*t* = 0 h) and after treatment (*t* = 24 h) (Supplementary Fig. 13a). Although this experimental system has been established previously[11], we validated the differential growth response by removing slowly the transparent tag from the touched leaf and showing that the hyponastic leaf remained upward with only a slight reduction of hyponasty (Supplementary Fig. 13c). All experiments started at 10:00 in the morning (ZT = 2 h). Brushing experiments were performed manually, trichomes of the fifth-youngest leaf were touched gently with a brush three times every 10 minutes for 7 hours until the photoperiod ended, starting at 10:00 in the morning (ZT = 2 h). Petiole angles were determined with image-J software directly at 0 and 7 hours of treatment. For competition assays 11 days old seedlings were transplanted to tree trays with individual pots (to avoid any root competition) to create dense stands of 8 × 8 plants (pots 19 ml and distance between the plants 2.2 cm). Col-0, L*er*, Fran-3, Wil-2, *gl1 (Ler background), ttg1, gl1* (Col-0 background), *glr3.1glr3.3glr3.6* and 9354 plants were transplanted to monoculture (Col-0 or 9354, Col-0 or *gl1*, Col-0 or *glr3.1glr3.3glr3.6*, Col-0 or Fran-3, Col-0 or Wil-2, L*er* or *gl1*, L*er* or *ttg1*) and mixed (Col-0 and 9354, Col-0 and *gl1*, Col-0 and *glr3.1glr3.3glr3.6*, Col-0 and Fran-3, Col-0 and Wil-2, L*er* and *gl1*, L*er* and *ttg1* in a 1:1 checkerboard grid[60]) canopies plots. Plants in the outer rows of the canopy served to minimize edge effects and only the 4 × 4 plants in the middle of the canopy were harvested when the canopy plots were 35 days old. Shoots were dried in an oven at 70 °C for 3 days and shoot dry weight of each individual plant was recorded.

**Far-red (FR) light treatments**
Supplemental FR light treatments were performed using a localized FR beam focused on the leaf tip (see Pantazopoulou et al.[10] for details) and

abbreviated as W+FR_tip. All growth conditions were standard as mentioned above, except that locally on the leaf tip R:FR dropped from 2.3 to 0.05. The R:FR treatment had no effect on the photosynthetically active radiation.

**Pharmacological experiments**
Plants were treated with different concentrations of the hormones MeJA (Van Meeuwen Chemicals BV, NL), or ABA (Sigma-Aldrich, USA). MeJA was given to the leaf blade, whereas ABA was applied to the petiole. All solutions, including the mock treatments, contained 0.1% DMSO and 0.1% Tween. The solutions were freshly made and they were applied right before and 5 h after the touch treatment. Mastoparan (Merck, NL) applications were performed right before the start of the hyponasty experiment. The solutions used for the experiments consisted of 40 μM Mastoparan dissolved in water with 0.1% Tween. The solution was freshly made for each experiment and a droplet of 4 μl was placed at three different locations, namely leaf tip, middle of the petiole, and petiole-lamina junction. Physiological experiments with LaCl₃ (Sigma-Aldrich, USA) and Verapamil (Merck, NL) treatment were done with application of the drug 24 h before (ZT = 2) the touch or W+FR_tip treatment. We used 2 mM of LaCl₃ and 1 mM of Verapamil freshly made for each experiment in water with 0.1% Tween, which was applied with a brush to the whole leaf. Two different concentrations of LaCl3 and Verapamil were to monitor the [Ca2 + ]cyt changes using the calcium biosensor *UBQ10p::GCaMP3* line: 2 mM and 50 mM for LaCl3, and 1 mM and 50 mM for Verapamil. The higher concentration was employed to ensure a rapid blockade of calcium. A softly applied droplet of 10 μl was placed at the leaf tip 25 minutes before the experiment, and then removed using Kimwipes papers to completely absorb it. The plants were allowed to recover for 5 minutes before the experiments commenced. For Mastoparan, a 40 μM solution was used, which was applied as a 10 μl droplet 2 minutes after monitoring the calcium visualization in Supplementary Fig. 6a. The solution was freshly made with water and 0.1% Tween.

**Transcriptome data analysis**
The transcriptome data on touch treatment were collected in a larger experiment that also included local FR treatments, and we published the FR transcriptome data in Pantazopoulou et al., 2017[10]. The harvesting, extraction, and processing as well statistical analyses are essentially as previously published[10]. In brief, the leaf tip and petiole base from wild-type plants (Col-0) were harvested after 5 h of touch

treatment, which is when leaf movement just becomes visible[11]. 15 petiole bases and 15 leaf tips were pooled for each sample for RNA extraction [three biological replicates (independent experiments) per tissue per treatment, collected from three independent experiments]. Affymetrix 1.1 ST *Arabidopsis* arrays were used to hybridize the samples via a commercial provider (Aros, Aarhus, Denmark). The raw data were normalized for signal intensity to remove background noise. The quality check of the data was performed using Bioconductor (packages "oligo" and "pd.aragene.1.1.st") in R software. Differential expression analysis was carried out using the Bioconductor "Limma" package in R software. Genes with adjusted $p$ value < 0,05 were considered as differentially expressed. Gene ontology (GO) analysis was done with GeneCodis (http://genecodis.cnb.csic.es). Clustering was based on the positive and negative logFC for each set. Genes with $p$ values ≤0.05 considered significant. The data are available in the National Center for Biotechnology Gene Expression Omnibus database (https://www.ncbi.nlm.nih.gov/geo/query/acc.cgi; accession no. GSE98643).

### GCaMP3 fluorescence visualization and quantification

The GCaMP3 fluorescence is quantified via the $\Delta F/F$ ratio. $\Delta F/F = (F - F_0)/F_0$, where $F$ is the GCaMP3 fluorescence of a given time point during touch while $F_0$ is the averaged based line in the ROIs in the first 2 min before touch treatment. The GCaMP3 fluorescence calculation for the leaf was performed in each selected leaf position for every ten seconds, while in the trichomes for every second. Video recordings were made with a 1.5× objective on an SMZ18 stereomicroscope (Nikon Instruments Europe BV, Amsterdam, Netherlands) equipped with an ORCA-Flash4.0 (C11440) camera (Hamamatsu, Solothurn, Switzerland) and eGFP emission/excitation filter set (AHF Analysentechnik AG, Tübingen, Germany). Light was supplied to the stereomicroscope using fiber optics. Video with a resolution of 512 × 512 pixels was acquired using NIS-Elements software (Nikon) with 1 frame s$^{-1}$ frequency. Video recordings for Figs. 5c–f, 6a, b, supplemental Figs. 5–7, 8e, f, and 9 were made with a 1x objective (total magnification ×0.77) on a M205FA stereomicroscope (Leica Microsystems BV Amsterdam, the Netherlands) and ET GFP band pass filter cube (Leica). Videos with a resolution of 1920 × 1440 pixels were acquired using LAS X (Leica) with 1 frame s$^{-1}$ frequency. Recordings were carried out in the dark at 22 °C.

### Statistical analysis

Data were analyzed with one or two-way ANOVA followed by Tukey's HSD test using GraphPad and Rstudio.

### Reporting summary

Further information on research design is available in the Nature Portfolio Reporting Summary linked to this article.

## Data availability

The authors declare that all the data supporting the findings of this study are available within this manuscript and Supplemental information files or available upon request to the corresponding authors. Transcriptomic data have been deposited in the National Center for Biotechnology Gene Expression Omnibus database (accession no. GSE98643). Source data are provided with this paper.

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

## Acknowledgements

We thank the Plant-Environment Signaling group (UU) for help with tissue harvests for the transcriptome experiments, Yorrit van de Kaa for preparing the plant material for the experiments, Debatosh Das and Ronnie de Jong for help with bioinformatics analysis, and Muthanna Biddanda Devaiah for helping with the photographic pictures. This work was funded by the Netherlands Organization for Scientific Research: Vidi 86512.003 (R.P. and C.K.P.), ENW Vici 865.17.002 (R.P.), and ALW open ALWOP.509 (C.K.P. and K.K.).

## Author contributions

C.K.P. and R.P. designed research, with additional input from E.E.F. and D.A.W.; C.K.P., S.B., C.T.N., and L.O. performed research; C.K.P., S.B., C.T.N., and K.K. analyzed data; C.K.P. wrote the manuscript draft; D.A.W., E.E.F., K.K., and R.P. revised the manuscript.

## Competing interests

The authors declare no competing interests.
