## [Peer Review File · Nature Communications]

REVIEWER COMMENTS

Reviewer #1 (Remarks to the Author):

Pantazopoulou et al. demonstrate in “Mechanodetection of neighbor plants elicits adaptive leaf movements through calcium dynamics” that touching trichomes of the leaf tip causes hyponasty through systemic calcium influx towards the petiole. Previously, Pantazopoulou et al. (2017) found that enrichment of FR light (low red/far-red) at the leaf tip induces hyponasty through auxin synthesis and transport in a PIF-dependent manner. Importantly, the authors of this study conclude that touch-induced hyponastic leaf movement is mechanistically distinct from FR light-mediated hyponasty.

The conclusion on the trichomes being important for perceiving neighbor plants is interesting, however, it is premature that this is a mechanism of adaptive strategy in dense vegetation.

Detailed comments:

1) The authors measured the petiole angles using a variety of mutant plants to evaluate the levels of hyponastic response. However, since hyponasty could be affected not only by the physiological properties of the petiole and leaf but also by physical characteristics, including petiole thickness and length, leaf weight, and cell wall stiffness, the comparisons of hyponastic responses between wild-type Col-0 and mutant plants should be carefully performed. For example, the *pin3pin4pin7* mutant displays severe developmental defects (Blilou et al., 2005, Nature), indicating that above mentioned physical traits of the triple mutant are different from those observed in Col-0. Then, although there is no significant difference in petiole angles between wild type and the triple mutant, how can we conclude that touch-induced hyponasty in the *pin3pin4pin7* mutant is qualitatively the same as in Col-0?

2) The same argument can be applied to the use of trichome mutants. The authors argue that trichomes are required for touch-induced hyponasty since the *gl1* and *ttg1* mutants show significantly reduced petiole angles. Notably, these mutants have defects in cuticle development (Xia et al., 2010, Plant Physiol.), and thus the physical properties of plant architecture in *gl1* plants are quite different from those of wild type. Therefore, I think it will be necessary to uncouple the effect of *gl1* on the trichomes versus the cuticle formation.

3) In the Methods section, the authors described that petiole angles to the horizontal were measured with ImageJ. How is the spatial orientation of the petiole determined for the angle measurement? This should be described and/or illustrated. Also, the images used for the angle measurement should be shown in the Figure and/or Supplementary Figure.

4) To induce touch-induced hyponasty, the authors used a transparent tag placed next to the fifth-youngest leaf for 24 h. It could be possible that, as the plants grow, the leaf leans on (or pushes) the transparent tag to angle the petiole independent of hyponasty. What happens if the tag is removed right before the measurement of petiole angles? Do the hyponastic leaf and petiole still maintain the angle?

5) The photo of Fig. 2a is quite similar to the one used in the previous PNAS paper (Fig. 3A; de Wit et al., 2012). I am not sure if a cropped version of an already published photo can be used for Nat Commun., so please check it.

6) The photo of Supplementary Fig. 1b shows *Nicotiana benthamiana* leaves inducing a hyponastic response against a transparent tag. Apparently, all the leaves seem to induce hyponastic upward movements. Is this a normal response in *N. benthamiana*? Please describe this in the main text and the difference compared to what is observed in *Arabidopsis thaliana*.

7) The authors concluded from the genetic analysis that the phytohormones, auxin, ABA, and JA are not involved in touch-induced hyponasty. Previous studies demonstrated that mechanostimulation induces these phytohormones as well as ethylene and reduces GA₄ levels. Indeed, the JA-responsive MYC mutant, *myc2myc3myc4* displayed a significant reduction in the petiole angle. Therefore, to validate the authors' conclusion, the quantification of phytohormones should be performed.

8) The authors performed transcriptome analysis to unveil the mechanism of touch-induced hyponasty. The leaf tip and petiole base were harvested 5 h after touch treatment for the expression analysis. How was the touch treatment performed? How did the authors determine the treatment time? Since only a minor number of DEGs were found by the authors' analysis, the timing of transcriptome analysis should be considered empirically.

9) The authors should perform in-depth expression analysis on both upregulated and downregulated genes independently to characterize hyponasty (Fig. 2). Also, the expression profiles should be compared with published datasets obtained by mechanostimulation (Van Moerkercke et al., PNAS, 2019; Xu et al., Plant J., 2019; Wang et al., PNAS, 2018). In addition, trichome-mediated expression profiles are available in Matsumura et al. (2021) paper (doi: <https://doi.org/10.1101/2021.06.13.448005>). In the current manuscript, the transcriptome analysis did not find a candidate signaling pathway, and thus mechanosensation-induced profiles may help the authors unveil the mechanism.

10) As far as I searched, there are many wrong annotations of DEGs (excel file) and thus the data cannot be evaluated correctly.

11) Both Pantazopoulou et al. and Matsumura et al. studies demonstrate that mechanostimulation of trichomes transiently induces calcium influx concentrically away from trichomes, however, I cannot see the GCaMP3 signal spread from trichomes to the petiole along the primary vein. In Fig. 3a and related video, I cannot recognize calcium signal initiated from trichomes rather the GCaMP3 signal seems to be generated via bending leaves. Therefore, it is not sure if trichome-induced calcium signal is involved in hyponasty. The GCaMP3 signal should be monitored in the *g1* and *glr* triple mutant backgrounds to link trichome-induced calcium to hyponasty.

12) Precise method is required for LaCl_3 treatment.

Reviewer #2 (Remarks to the Author):

In this paper by Pantazopoulou et al, the underlying signalling that controls how plants sense their neighbours through touch sensitivity is explored. The work is an original take on this interesting field and provides important new insights. The paper is clearly written and easy to follow up. Most of the points are clearly made, but some additional validations and examinations would lift the quality of the work further.

- The authors start with a transcriptomic study comparing changes in leaf tips vs petioles in 'tip-touched' leaves. This is an interesting approach, but I feel the data is a bit under-explored, especially in terms of how this neighbour detection at both sites compares to other touch and water spray-induced transcriptomes that were published in the last years. It would be interesting to know if both expression patterns (tip vs petiole) are observed when whole leaves are touched, or if it is e.g. more similar to only the tip transcriptome. Also it would be good to provide some list of genes in the manuscript to see exactly what is happening, besides just general GO terms. For instance the top 'tip' or 'petiole' genes, or the overlap of 35 genes could be mentioned in a figure or table in the main manuscript.

- The role of JA is suggested by the transcriptome data in the tip, suggesting that JA is released in the tip area that may start the signalling. The *myc234* mutant data at least partially support that concept. I was intrigued by the experiment where MeJA was added to the leaf tip to check petiole angle changes (Suppl Fig 3A). There seems to be some induction in the angle at the higher concentration, but that is apparently not significant. It would be interesting to see if this experiment can be expanded, perhaps with some higher concentrations. I would like to see if the JA itself can trigger the bending or if is just a side effect and the Ca^{2+} is the main signal.

- Line 123: I am wondering where the statement that MeJA can limit petiole growth and thereby hyponasty comes from. There seems to be some evidence presented for ABA in Fig S3 f, but I don't see what the MeJA statement is based on.

- It is interesting to see that the far-red induced hyponasty is not lost in *glr3* triple mutants (Fig 4c), only touch-related hyponasty. I am wondering what happens in the *myc234* mutant under FR treatment, as there is some evidence that JA is linked to light-regulated shade avoidance.

- The graphs in figure 3c should be analysed statistically, to see e.g. after which time the difference between touch and control occurs at the various distances

- Fig 5 holds some interesting data on how leaf tip touch sensing via trichomes is important for regulating competition between neighbouring plants. The pictures in Figure 5g are interesting, but it would be good to get some more detailed insights. Can differences in Col-0 and 9354 be observed in terms of hyponastic growth in mixed culture? Do the Col-0 grow over the 9354? It would be interesting to know if a similar loss of competitiveness can be observed in FR signalling (e.g. *pif* mutants) or *glr* triple mutants.

- The video's illustrate nicely the Ca^{2+} wave observed around the trichomes. In Video 1 the touched leaf tip does show some kind of Ca^{2+} transients as the authors suggest. To me, there seem to be two (or even more) waves: one or more early ones, and then a stronger one towards the end. Such a double peak can also be somewhat observed in Fig 3C. Is this reproducible and can the authors comments on this?

- Trichomes undoubtedly help plants with amplifying mechanical cues, and the authors present supporting data the *gl1* glabrous mutants do not show as much hyponasty after touching (Fig 5e). Chehab et al., 2012 previously showed the role of JA synthesis in touch-induced resistance and morphology using the *aos* mutant, which is interestingly also in the *gl1* background. In their study the *gl1* mutant was used as a control for touch-induced thigmomogenesis/resistance compared to the *aos* (*gl1*) mutant and showed clear touch induced effects. This should be discussed and compared to the authors' currently presented work. These findings to me indicate that trichomes perhaps amplify the signal as physical extensions that provide extra leverage on the surrounding cells, but are not per se absolutely required for the response. Therefore, statements in the abstract and results that trichomes are required should be more nuanced. As the genetic background of the 9354 ecotype are not described or known, there may be other underlying mutations besides lack of trichomes that affect the mixed growth competition assays. As *gl1* is genetically more defined, such an experiment would be better than in the 9354 line.

Minor comments:

Line 15: elicit in: in should be removed

Line 53: should be 'induce'

The authors should consider moving suppl fig 5a-d to the main manuscript

Is the genetic basis of glabrous accessions such as 9354 known? If so, this is worth discussing.

Reviewer #3 (Remarks to the Author):

This manuscript describes how rosette plants may detect near neighbors through a mechanical interaction resulting in reorientation of leaves similar to leaf position changes that occur in response to shading. Strong genetic evidence is provided that indicates that the touch-induced hyponastic leaf movement response occurs through a pathway that is distinct from the better understood light-induced shade avoidance response. Possible fitness implications are provided.

The experiments described seek to provide insight into the mechanistic pathway and relevance of the touch-induced hyponastic leaf response. Cytoplasmic calcium signaling, GLR channels, and trichome structures are all implicated. Finally, a high-density growth competition assay provides some evidence that the touch-induced hyponastic leaf response may be beneficial.

The data presented are intriguing in that they indicate that plants may use touch responses to outcompete neighbors and implicate trichomes as being critical as sensory organs. However, additional experimental detail and data would strengthen confidence in the results and the overall conclusions.

First, it is not clear to this reviewer how the touch treatment is done. The methods describe that insertion of a transparent label into the soil adjacent to the 5th rosette leaf provides the stimulus. But since the time of the touch seems to be universal (ZT = 2hr), it seems that the insertion of the tag occurs in such a way to immediately touch the 5th leaf instead of letting the leaf gradually elongate leading to contact with the tag, which may more closely resemble what occurs during rosette expansion leading to neighbor detection. Exactly where the tag is placed relative to the leaf tip could differentially affect the force the leaf tip experiences. How do the researchers control for the level of force the leaf tip experiences to ensure uniformity and relevance to the force that might be expected from neighboring leaves?

Although the researchers intriguingly implicate Ca²⁺ signaling, GLR functions and trichomes in the touch-induced hyponasty response, the data for each of these is incomplete.

(1.) The Ca²⁺ reporter fluorescence increases with mechanical touch, consistent with what is known about touch-induced calcium changes in plants. The authors attempt to show a requirement for Ca²⁺ by treating plants with LaCl₃; LaCl₃ treatment reduced touch-induced hyponasty. However, they did not test whether the LaCl₃ treatment changed the Ca²⁺ reporter fluorescence changes in the leaf tip (and

trichomes) in response to touch or whether LaCl₃ altered the downstream changes in (at least a subset of) transcripts either at the tip or the base.

(2.) Similarly, the authors suggest that GLRs are part of the touch-induced hyponasty pathway because the mutants showed a reduced response, but did not include testing these mutants for changes in touch-induced Ca²⁺ fluorescence in leaf regions or trichomes, gene expression changes, hyponasty in dense growth conditions, or loss of competition with wild type.

(3.) The authors suggest that trichomes are important for touch-induced hyponasty using multiple mutants and accessions that lack trichomes, but only conducted more focused competition studies on one mutant. All of the mutants should be examined; otherwise if only the 9354 mutant is affected, then something specific about this mutant and not the loss of trichomes would be implicated as essential for touch-induced hyponasty. Furthermore all of the trichome mutants should be tested for changes in touch-induced Ca²⁺ fluorescence changes and gene expression changes to try to elucidate pathway components.

Other comments/questions:

Figure 1. labels and legend are inconsistent. Also, Box and whisker plots may be better to show individual variation in responses

Figure 2. Was a fold-change cut off used to identify transcript abundances?

Figure 5 shows that the Ca²⁺ reporter fluorescence is higher in trichomes than surrounding tissue. Could this be because trichomes have a higher concentration of the reporter or because the narrow translucent trichome structure makes it easier to detect fluorescence?

Title: To say that the leaf movements occur through calcium dynamics, additional data must be obtained. Although the authors show that touch stimulation causes a calcium change, they do not show that this calcium change is necessary or sufficient for the response. A non-specific calcium perturbator, LaCl₃, affects the growth response, but no evidence is provided to show that this treatment blocked the calcium signal change. Ideally, additional calcium perturbations would also be tested to rule out nonspecific effects. Finally, one must show that driving a calcium flux (in the absence of the touch stimulus) is sufficient to cause the downstream response. Otherwise, one can only emphasize a correlation of the response with calcium flux.

Abstract: Based on data presented, it is too strong to say that perturbation of the calcium response and the absence of trichomes “inhibit” touch-induced hyponasty.

Methods: What was the rationale to choose ZT=2hr? Is there a role for the circadian clock?

REVIEWER COMMENTS-rebuttal

Reviewer #1 (Remarks to the Author):

Pantazopoulou et al. demonstrate in “Mechanodetection of neighbor plants elicits adaptive leaf movements through calcium dynamics” that touching trichomes of the leaf tip causes hyponasty through systemic calcium influx towards the petiole. Previously, Pantazopoulou et al. (2017) found that enrichment of FR light (low red/far-red) at the leaf tip induces hyponasty through auxin synthesis and transport in a PIF-dependent manner. Importantly, the authors of this study conclude that touch-induced hyponastic leaf movement is mechanistically distinct from FR light-mediated hyponasty. The conclusion on the trichomes being important for perceiving neighbor plants is interesting, however, it is premature that this is a mechanism of adaptive strategy in dense vegetation.

Detailed comments:

1) The authors measured the petiole angles using a variety of mutant plants to evaluate the levels of hyponastic response. However, since hyponasty could be affected not only by the physiological properties of the petiole and leaf but also by physical characteristics, including petiole thickness and length, leaf weight, and cell wall stiffness, the comparisons of hyponastic responses between wild-type Col-0 and mutant plants should be carefully performed. For example, the *pin3pin4pin7* mutant displays severe developmental defects (Blilou et al., 2005, Nature), indicating that above mentioned physical traits of the triple mutant are different from those observed in Col-0. Then, although there is no significant difference in petiole angles between wild type and the triple mutant, how can we conclude that touch-induced hyponasty in the *pin3pin4pin7* mutant is qualitatively the same as in Col-0?

Blilou et al. 2005 showed that the *pin3pin4pin7* mutant had a reduction only at the elongation zone of the root (but not in the root or cell length and meristem zone). However, no data are available about the shoot of the adult plants in this paper. This *pin3pin4pin7* line has been previously used in our research (Pantazopoulou et al., 2017, PNAS) and others (Michaud et al., 2017, PNAS) without observing severe developmental defects. Although it has a FR-response defect, we show here that it is a wildtype-like response to touch treatment. We performed some additional analyses on the *pin3pin4pin7* mutants, focusing on the leaf that we typically do all hyponasty and petiole elongation experiments on. We observed that petiole length in *pin3pin4pin7* is only very mildly reduced compared to Col-0 (4,6 mm in the mutant compared to 5.1 mm in Col-0, figure R1 a & b, below). We also measured the fresh and dry weight and found no differences between the two genotypes (figure R1 c & d). Furthermore, we sectioned the petiole of this leaf in three parts, the petiole-lamina junction part of the petiole, the middle of the petiole and the basal part of the petiole close to the meristem, and measured the petiole width and thickness in all these different parts (figure e & f). We found no significant differences in any of the measurements of the petiole width and

thickness except of a small difference in the petiole-lamina junction part of the petiole, indicating that Col-0 is slightly thicker at this part than *pin3pin4pin7* (figure R1 e). However, it is worth mentioning that mutants that have similar growth rate as Col-0 for example *glr3.1glr3.3glr3.6* showed a reduced touch-induced hyponasty while *pin3pin4pin7* with a slight reduction showed a wild-type response. We propose that petiole length is not likely a major factor determining touch-induced hyponasty.

Figure R1: Col-0 and *pin3pin4pin7* growth data. (a) Illustration of Col-0 and *pin3pin4pin7* rosette form the top and side. (b) lamina and petiole length, (c) fresh weight, (d) dry weight, (e) petiole thickness and (f) petiole width of the fifth leaf of 28 days old Col-0 and *pin3pin4pin7* plants growing in control conditions. Data represent mean \pm SE; n = 10-15. Different letters or asterisks indicate significant differences (two-way ANOVA with Tukey's post hoc test or t-test; P < 0.05). (b) Length was measured with a caliper while (e,f) petiole thickness and width via image j. (c,d) Fresh and dry weigh was measured with a four digital scale. The fifth youngest leaf of 28 days old Col-0 and *pin3pin4pin7* stayed to the 70 degrees oven for 3 days before we measured the dry weight.

2) The same argument can be applied to the use of trichome mutants. The authors argue that trichomes are required for touch-induced hyponasty since the *gl1* and *ttg1* mutants show significantly reduced petiole angles. Notably, these mutants have defects in cuticle development (Xia et al., 2010, Plant Physiol.), and thus the physical properties of plant architecture in *gl1* plants are quite different from those of wild type. Therefore, I think it will be necessary to uncouple the effect of *gl1* on the trichomes versus the cuticle formation. accessions

Indeed, based on the Xia et al., (2010) *gl1* and *ttg1* in Col-0 background show some defects in cuticle formation that compromise the defense of the plant to pathogens that may breach the cuticle. However, keep in mind that in our study there is no damage or wounding, since the leaf-leaf touching is a soft mechanostimulation as demonstrated by our calcium data as well. Furthermore, all the lines we used without trichomes (mutants and accessions) respond

to local FR-induced (W+FR_{tip}) hyponasty, indicating that there certainly is no basic mechanical problem with upward leaf movement in these lines. However, to verify if cuticle defects could affect hyponasty, we tested several mutants that have trichomes but express an impaired cuticle formation, *cer1*, *cer3* in Col-0 and *cer1* and *cer3* in *Ler* background. We found that all five cuticle formation mutants showed a wild-type-like touch-induced hyponasty (figure s10 f,g). These data indicate that cuticle defects do not necessarily affect touch-induced hyponasty, supporting the conclusion that *gl1* and *ttg* mutants have reduced touch-induced hyponasty because of their lack of trichomes.

3) In the Methods section, the authors described that petiole angles to the horizontal were measured with ImageJ. How is the spatial orientation of the petiole determined for the angle measurement? This should be described and/or illustrated.

We have now included an illustration in supplemental figure 13 to show this.

Also, the images used for the angle measurement should be shown in the Figure and/or Supplementary Figure.

We have also included extra example images for some of the experiments, illustrating the leaf movement responses as shown quantitatively in the graphs. Please see supplemental figure 6, 10, 11, 12.

4) To induce touch-induced hyponasty, the authors used a transparent tag placed next to the fifth-youngest leaf for 24 h. It could be possible that, as the plants grow, the leaf leans on (or pushes) the transparent tag to angle the petiole independent of hyponasty. What happens if the tag is removed right before the measurement of petiole angles? Do the hyponastic leaf and petiole still maintain the angle?

Great question. We did exactly the experiment suggested and have included the data in supplemental figure 13. What we observed is that the touch-triggered hyponastic leaves indeed stay elevated upon removal of the tag, but slightly reduced as compared to before tag removal. Triggered by a comment from one of the other reviewers we also found that just triggering a calcium increase with a drug called Mastoparan on the leaf tip induces hyponasty without any tag present (supplemental figure 6a,b), further indicating that this response does not depend on leaning on a tag.

Zooming out and considering the real plant competition setting: When two leaves are touching in a dense canopy they will remain touched at least for the first 24h, often longer. We showed previously (de Wit et al 2012, PNAS) that the touch-induced hyponastic plants will create horizontal far-red light reflection, consolidating the shade avoidance phenotypes and further promoting hyponasty beyond the touch-induced response magnitude.

5) The photo of Fig. 2a is quite similar to the one used in the previous PNAS paper (Fig. 3A; de Wit et al., 2012). I am not sure if a cropped version of an already published photo can be used for Nat Commun., so please check it.

We would like to thank the reviewer for noticing! This was a mix-up from our side. We have replaced the figure with a new one.

6) The photo of Supplementary Fig. 1b shows *Nicotiana benthamiana* leaves inducing a

hyponastic response against a transparent tag. Apparently, all the leaves seem to induce hyponastic upward movements. Is this a normal response in *N. benthamiana*? Please describe this in the main text and the difference compared to what is observed in *Arabidopsis thaliana*.

We had not measured a putative systemic response in other species except *Arabidopsis*. To verify if touch does or does not induce a systemic response in *Nicotiana benthamiana* we performed a new experiment in which we measured the systemic responses (petiole angle) in this species as well. No systemic effects were found at all (supplemental figure 1b,d), and we have replaced the original photo, that was taken from an angled perspective and gave the impression of systemic responses, with a new one (supplemental figure 1c).

7) The authors concluded from the genetic analysis that the phytohormones, auxin, ABA, and JA are not involved in touch-induced hyponasty. Previous studies demonstrated that mechanostimulation induces these phytohormones as well as ethylene and reduces GA₄ levels. Indeed, the JA-responsive MYC mutant, *myc2myc3myc4* displayed a significant reduction in the petiole angle. Therefore, to validate the authors' conclusion, the quantification of phytohormones should be performed.

Indeed, *myc2myc3myc4* displayed a slight reduction in touch-induced hyponasty but to the same extent as *myc2*. Interestingly none of the JA biosynthesis and receptor mutants showed any reduced hyponastic responses due to touch. Since MYC2 is involved in various pathways, including light responses, its modestly affected hyponasty is not straightforwardly interpreted. Regarding ABA, all the ABA biosynthesis perception and signaling mutants showed wild-type-like touch-induced hyponasty (supplemental figure 4i-k). Based on this we do not expect major changes in these hormones between touch and control conditions. Furthermore, in line 131 we conclude that these two hormones do not play a major role in touch induced hyponasty but rather denote a mechanostimulation response.

As agreed in consultation with the editor, we have not followed up on hormone measurements since they are not likely causally related to the response studied here.

8) The authors performed transcriptome analysis to unveil the mechanism of touch-induced hyponasty. The leaf tip and petiole base were harvested 5 h after touch treatment for the expression analysis. How was the touch treatment performed? How did the authors determine the treatment time? Since only a minor number of DEGs were found by the authors' analysis, the timing of transcriptome analysis should be considered empirically.

Based on the time-course of hyponasty (de Wit et al., 2012, PNAS), the first notable upward movement of the leaf was observed 5h after start of touch treatment, and we chose to use this timepoint for our harvests, as is now explained in the materials and methods section (line 431-432). The touch treatment was performed with a transparent tag as in most other experiments.

9) The authors should perform in-depth expression analysis on both upregulated and downregulated genes independently to characterize hyponasty (Fig. 2). Also, the expression profiles should be compared with published datasets obtained by mechanostimulation (Van Moerkercke et al., PNAS, 2019; Xu et al., Plant J., 2019; Wang et al., PNAS, 2018). In addition, trichome-mediated expression profiles are available in Matsumura et al. (2021) paper (doi: <https://doi.org/10.1101/2021.06.13.448005>). In the current manuscript, the

transcriptome analysis did not find a candidate signaling pathway, and thus mechanosensation-induced profiles may help the authors unveil the mechanism.

Thanks for these suggestions. We now include the relevant comparisons of our transcriptome analysis against the suggested papers now included as supplemental figure 3). Since many of the up and down-regulated genes have unknown functions, an in-depth analysis is difficult. Regardless, the low number of differentially expressed genes in our samples indicates that this is a very mild treatment that is not directly comparable to more severe mechanical perturbations.

10) As far as I searched, there are many wrong annotations of DEGs (excel file) and thus the data cannot be evaluated correctly.

We would like to thank the reviewer for this feedback. We carefully checked the annotations and updated them in the excel file.

11) Both Pantazopoulou et al. and Matsumura et al. studies demonstrate that mechanostimulation of trichomes transiently induces calcium influx concentrically away from trichomes, however, I cannot see the GCaMP3 signal spread from trichomes to the petiole along the primary vein. In Fig. 3a and related video, I cannot recognize calcium signal initiated from trichomes rather the GCaMP3 signal seems to be generated via bending leaves. Therefore, it is not sure if trichome-induced calcium signal is involved in hyponasty. The GCaMP3 signal should be monitored in the *gl1* and *glr*triple mutant backgrounds to link trichome-induced calcium to hyponasty.

These are very good suggestions that took us quite a bit of time since some new genetic crosses had to be made. We performed the following experiments:

1) use the *UBQ10::GCaMP3* line to touch the trichomes of the leaf tip close to the main vein. This experiment showed that calcium is spreading from the trichome to the primary vein (figure 5c,d and supplemental figure 8e).

2) We made a cross between *UBQ10::GCaMP3* x *gl1* and recorded the calcium dynamics upon touch between wild-type and *gl1* genotype expressing the calcium sensor. The data revealed a strong impairment of the touch-induced cytosolic calcium induction in the *gl1* mutants as compared to wild-type (figure 6a,b and supplemental figure 9).

3) We also crossed the *glr3.1glr3.3glr3.6* triple mutant with the *UBQ10::GCaMP3* line and recorded the calcium dynamics upon touch. We found that in the *glr3.1glr3.3glr3.6* with *UBQ10::GCaMP3* the GCaMP3 fluorescence is strongly reduced as compared to this sensor in the wild-type background (figure 4d and supplemental figure 7).

12) Precise method is required for LaCl_3 treatment.

We added more detailed of how we performed the LaCl_3 experiments in Materials and methods (lines 333-339).

Reviewer #2 (Remarks to the Author):

In this paper by Pantazopoulou et al, the underlying signalling that controls how plants sense their neighbours through touch sensitivity is explored. The work is an original take on this interesting field and provides important new insights. The paper is clearly written and easy to follow up. Most of the points are clearly made, but some additional validations and examinations would lift the quality of the work further.

1-The authors start with a transcriptomic study comparing changes in leaf tips vs petioles in 'tip-touched' leaves. This is an interesting approach, but I feel the data is a bit under-explored, especially in terms of how this neighbour detection at both sites compares to other touch and water spray-induced transcriptomes that were published in the last years. It would be interesting to know if both expression patterns (tip vs petiole) are observed when whole leaves are touched, or if it is e.g. more similar to only the tip transcriptome. Also it would be good to provide some list of genes in the manuscript to see exactly what is happening, besides just general GO terms. For instance the top 'tip' or 'petiole' genes, or the overlap of 35 genes could be mentioned in a figure or table in the main manuscript.

We compared our data set with the previously published transcriptomic data, please see reviewer 1 comment 9. Also, the 35 and 15 overlap genes in leaf tip and petiole base respectively for supplemental figure 2 can be found in the excel file 1_DEGS with all the details (name of the gene, gene assign and logFC).

2-The role of JA is suggested by the transcriptome data in the tip, suggesting that JA is released in the tip area that may start the signalling. The myc234 mutant data at least partially support that concept. I was intrigued by the experiment where MeJA was added to the leaf tip to check petiole angle changes (Suppl Fig 3A). There seems to be some induction in the angle at the higher concentration, but that is apparently not significant. It would be interesting to see if this experiment can be expanded, perhaps with some higher concentrations. I would like to see if the JA itself can trigger the bending or if is just a side effect and the Ca²⁺ is the main signal.

We performed a MeJA experiment in higher concentration (200µM MeJA) and we observed strong reduction in the hyponastic responses and petiole elongation (supplemental figure 4a) rather than induction, confirming that JA is not likely a positive regulator of touch-induced hyponasty.

3-Line 123: I am wondering where the statement that MeJA can limit petiole growth and thereby hyponasty comes from. There seems to be some evidence presented for ABA in Fig S3 f, but I don't see what the MeJA statement is based on.

We were not entirely clear in our writing, but now that we have performed the experiment suggested by the reviewer, the data are in the manuscript. These findings are consistent with findings on JA-induced leaf growth inhibition in for example Yan et al., 2007, Plant Cell and Zhang et al., 2008, PLoS ONE.

4-It is interesting to see that the far-red induced hyponasty is not lost in glr3 triple mutants (Fig 4c), only touch-related hyponasty. I am wondering what happens in the myc234

mutant under FR treatment, as there is some evidence that JA is linked to light-regulated shade avoidance.

Indeed, some links exist and we performed the requested experiment: we found that FR treatment of the leaf tip induced a similar hyponastic response in Col-0 wild-type and the *myc2myc3myc4* triple mutant (supplemental figure 4f).

5- The graphs in figure 3c should be analysed statistically, to see e.g. after which time the difference between touch and control occurs at the various distances

We had already performed a statistical analysis in all the six different positions (table 1). The comparisons have been performed between 0-500 sec and 500-1000 sec and showed a significant difference in all the positions between the two different time frames except position 6 between 0-500 sec. In order to find at which time, the difference between touch and control occurs we refined the window and checked between 0-200 sec and 200-500 sec in all the positions (i.e when the calcium signal starts to increase). Time since start of the experiment is a significant factor for the calcium increase the first 200 sec but the significant interaction between treatment and time is between 200-500 sec (position 1 to 4). For more details please see table 1.

6- Fig 5 holds some interesting data on how leaf tip touch sensing via trichomes is important for regulating competition between neighbouring plants. The pictures in Figure 5g are interesting, but it would be good to get some more detailed insights. Can differences in Col-0 and 9354 be observed in terms of hyponastic growth in mixed culture? Do the Col-0 grow over the 9354? It would be interesting to know if a similar loss of competitiveness can be observed in FR signalling (e.g. *pif* mutants) or *glr* triple mutants.

To answer the question if Col-0 can overtop the 9354 accession, we performed an experiment with Col-0 and 9354 growing side by side in pairs and observed that Col-0 indeed typically overlapped 9354. The same experiment was performed with all the lines used here that lack trichomes and with *glr3.1glr3.3glr3.6* against the wild-type and the results were similar to those found for the Col-0 versus 9354 overlapping experiment (see supplemental figure 12c). As for FR signaling mutants, we have published competition experiments previously. Indeed, Keuskamp et al., 2010 (PNAS) tested a mixed canopy with Col-0 (responds to far-red) and *pin3* (does not respond to far-red) plants and observed that indeed *pin3* lost the competition against Col-0 wildtype indeed. Also, *pif4pif5pif7* (unresponsive to far-red and shade) growing in Col-0 canopy, was severely suppressed by Col-0 wildtype (*Pantazopoulou et al., 2020, Plant Cell and Environment*).

7- The video's illustrate nicely the Ca²⁺ wave observed around the trichomes. In Video 1 the touched leaf tip does show some kind of Ca²⁺ transients as the authors suggest. To me, there seem to be two (or even more) waves: one or more early ones, and then a stronger one towards the end. Such a double peak can also be somewhat observed in Fig 3C. Is this reproducible and can the authors comments on this?

We would like to thank the reviewers for noticing it. We checked the individual data of the touch treatment to see the reproducibility. Indeed, we noticed two calcium waves at the

leaf tip (which is the treated side) in the majority of the plants and a very few times three. We have included this observation in the manuscript, lines 153-154.

8- Trichomes undoubtedly help plants with amplifying mechanical cues, and the authors present supporting data the *gl1* glabrous mutants do not show as much hyponasty after touching (Fig 5e). Chehab et al., 2012 previously showed the role of JA synthesis in touch-induced resistance and morphology using the *aos* mutant, which is interestingly also in the *gl1* background. In their study the *gl1* mutant was used as a control for touch-induced thigmomogenesis/resistance compared to the *aos* (*gl1*) mutant and showed clear touch induced effects. This should be discussed and compared to the authors' currently presented work.

The *aos* line we used in the manuscript has trichomes. The *aos* line was backcrossed to Col-0 and the trichomes phenotype was restored. This line was published in Yan et al., 2007 Plant Cell. Indeed, in our previous paper, the *aos* mutant in the *gl1* background displayed a reduced touch-induced hyponasty (de Wit et al., 2012, PNAS), but the *aos* in wildtype Col-0 background showed a wild-type induced hyponasty (this manuscript). These data indicate that the phenotype is due to the trichomes and not due to *aos* function.

Please note: The touch responses of Chehab et al., 2012 are quite different to our set-up. Chehab et al., 2012 used a gentle touch by hand to bend the leaves up and down 10 times, while in our case is a continues touch for 24h. Also, the measurements are different since we measure hyponastic responses while they measured % of flowering and lesion size upon touch. All these different factors can lead to different results as we also showed based on the comparative analysis we did with other mechanostimulation studies and our data set in supplemental figure 2.

9-These findings to me indicate that trichomes perhaps amplify the signal as physical extensions that provide extra leverage on the surrounding cells, but are not per se absolutely required for the response. Therefore, statements in the abstract and results that trichomes are required should be more nuanced. As the genetic background of the 9354 ecotype are not described or known, there may be other underlying mutations besides lack of trichomes that affect the mixed growth competition assays. As *gl1* is genetically more defined, such an experiment would be better than in the 9354 line.

-Taking into consideration this reviewer comment we nuanced the statements in lines 20, 6264 and 245.

-Bloomer et al., 2019, Molecular Ecology showed that 9354 has a point mutation which leads in this glabrous phenotype. To make sure that indeed trichomes are important in the touch-induced hyponasty, we tested several lines without trichomes as you can see in Figure 6d,e and supplemental figure 10a (we add the genetic info in material and methods section). Moreover, according to reviewer's comment, we included the *gl1* in Col-0 background and performed this experiment and found strong reduction in the touch-induced hyponasty as the previously tested glabrous line.

Minor comments:

Line 15: elicit in: in should be removed

We removed “in” from the text line 14

Line 53: should be 'induce'

We corrected it in line 54.

The authors should consider moving suppl fig 5a-d to the main manuscript

We did move the gl1 in Col-0 background to the main figure. Since the other accession is an extra confirmation of our findings we suggest keeping it in the supplements.

Is the genetic basis of glabrous accessions such as 9354 known? If so, this is worth discussing.

we add it in material and methods section lines 283-284

Reviewer #3 (Remarks to the Author):

This manuscript describes how rosette plants may detect near neighbors through a mechanical interaction resulting in reorientation of leaves similar to leaf position changes that occur in response to shading. Strong genetic evidence is provided that indicates that the touch-induced hyponastic leaf movement response occurs through a pathway that is distinct from the better understood light-induced shade avoidance response. Possible fitness implications are provided.

The experiments described seek to provide insight into the mechanistic pathway and relevance of the touch-induced hyponastic leaf response. Cytoplasmic calcium signaling, GLR channels, and trichome structures are all implicated. Finally, a high-density growth competition assay provides some evidence that the touch-induced hyponastic leaf response may be beneficial.

The data presented are intriguing in that they indicate that plants may use touch responses to outcompete neighbors and implicate trichomes as being critical as sensory organs. However, additional experimental detail and data would strengthen confidence in the results and the overall conclusions.

First, it is not clear to this reviewer how the touch treatment is done. The methods describe that insertion of a transparent label into the soil adjacent to the 5th rosette leaf provides the stimulus. But since the time of the touch seems to be universal (ZT = 2hr), it seems that the insertion of the tag occurs in such a way to immediately touch the 5th leaf instead of letting the leaf gradually elongate leading to contact with the tag, which may more closely resemble what occurs during rosette expansion leading to neighbor detection. Exactly where the tag is placed relative to the leaf tip could differentially affect the force the leaf tip experiences. How do the researchers control for the level of force the leaf tip experiences to ensure uniformity and relevance to the force that might be expected from neighboring leaves?

Please see our response to reviewer 1 comment 3

Although the researchers intriguingly implicate Ca²⁺ signaling, GLR functions and trichomes in the touch-induced hyponasty response, the data for each of these is incomplete.

(1.) The Ca²⁺ reporter fluorescence increases with mechanical touch, consistent with what is known about touch-induced calcium changes in plants. The authors attempt to show a requirement for Ca²⁺ by treating plants with LaCl₃; LaCl₃ treatment reduced touch-induced hyponasty. However, they did not test whether the LaCl₃ treatment changed the Ca²⁺ reporter fluorescence changes in the leaf tip (and trichomes) in response to touch or whether LaCl₃ altered the downstream changes in (at least a subset of) transcripts either at the tip or the base.

Great remark and in order to make this aspect more solid, we chose not to focus on gene expression but to perform the proposed experiment with the GCaMP3 calcium sensor that is a direct and functionally relevant read-out for mechanostimulation response. We applied LaCl₃ on the leaf tip of the *UBQ10::GCaMP3* line and monitored the calcium dynamics, observing a strong reduction of the calcium signal upon exogenous application of LaCl₃, see Supplemental figure 5a-g. Furthermore, to corroborate the LaCl₃ findings, we decided to use a second inhibitor, Verapamil and we found similar results (see Supplemental figure 5h-n for

the calcium signal and supplemental figure 5o,p for the hyponastic response). These data corroborate our initial findings on the requirement of Ca²⁺ for touch-induced hyponasty.

(2.) Similarly, the authors suggest that GLRs are part of the touch-induced hyponasty pathway because the mutants showed a reduced response, but did not include testing these mutants for changes in touch-induced Ca²⁺ fluorescence in leaf regions or trichomes, gene expression changes, hyponasty in dense growth conditions, or loss of competition with wild type.

-Please see reviewer 1 comment 11 regarding the Ca²⁺ fluorescence in leaf regions: we made a combined *glr3.1glr3.3glr3.6 X GCaMP3* line through crossing, which confirmed GLR involvement in propagation the Ca²⁺ wave.

-regarding competition experiments in dense growth conditions: we decided to perform a competition experiment at high density between Col-0 and *glr3.1glr3.3glr3.6* and compared these to their performance in their respective monocultures. We found a reduction in the dry weight of the *glr3.1glr3.3glr3.6* triple mutant, confirming a selective disadvantage of this genotype against wild-type. This is consistent with the interpretation that the lack of hyponasty due to touch in *glr3.1glr3.3glr3.6* gives an advantage to the well-responding Col-0 wild-type that overlaps early on in canopy development leading to superior performance of wildtype against *glr3.1glr3.3glr3.6* mutant (Supplemental figure12b,c).

(3.) The authors suggest that trichomes are important for touch-induced hyponasty using multiple mutants and accessions that lack trichomes, but only conducted more focused competition studies on one mutant. All of the mutants should be examined; otherwise if only the 9354 mutant is affected, then something specific about this mutant and not the loss of trichomes would be implicated as essential for touch-induced hyponasty.

Although competition assays are very time- and space-consuming (hence having limited them initially) we understand this specific request because 9354 is an accession and may thus carry other genetic alterations as compared to Col-0 than just the *gl1* mutation. We therefore performed competition assays with all the mutants that showed defective touch-induced hyponasty. All of them showed a dry weight reduction during competition with their respective wild-type Col-0 or *Ler* (figure 7 and supplemental figure 11 and 12). This indicates that it is not specific to the 9354 mutation, but a general pattern of glabrous genotypes.

Furthermore all of the trichome mutants should be tested for changes in touch-induced Ca²⁺ fluorescence changes and gene expression changes to try to elucidate pathway components.

We understand that it is always interesting to study every question in every line, and cross the reporter to every mutant at hand. Although that in its entirety was not feasible, we did study the key element that is requested here: the *gl1* glabrous mutant crossed to the UBQ10:GCaMP3 reporter as well as the *glr3.1glr3.3glr3.6* crossed to the UBQ10:GCaMP3 reporter. We have also compared hyponastic responses of a variety of glabrous accessions and mutants, showing that all of them have strongly disturbed touch-induced hyponasty, making it very unlikely that responses observed in *gl1* are a side-effect of this specific mutation. We hope that our efforts that delivered a wealth of new data towards the reviewer's suggestion will be appreciated and considered of sufficient gravity to move forward with this submission.

Other comments/questions:

Figure 1. labels and legend are inconsistent.

We did check it and made all the corrections.

Also, Box and whisker plots may be better to show individual variation in responses

We kept the original bar graphs but have added the individual data points on top in order to see exactly what is requested: the individual variation.

Figure 2. Was a fold-change cut off used to identify transcript abundances?

We add it in the Material & Method section in line 364

Figure 5 shows that the Ca²⁺ reporter fluorescence is higher in trichomes than surrounding tissue. Could this be because trichomes have a higher concentration of the reporter or because the narrow translucent trichome structure makes it easier to detect fluorescence?

Interestingly, the apparent output of a number of reporters appears to be particularly high in trichomes (e.g. iGluSnFR in Grenzi et al., 2023). It is most likely that this is caused by trichome translucency, and possibly a low level of autofluorescence. Measurements of GCaMP3 fluorescence is quantified via the $\Delta F/F$ ratio, which represents the relative change in fluorescence. Therefore, any higher initial signals are accounted for and should not compound the interpretation of figure 5.

Title: To say that the leaf movements occur through calcium dynamics, additional data must be obtained. Although the authors show that touch stimulation causes a calcium change, they do not show that this calcium change is necessary or sufficient for the response. A non-specific calcium perturbator, LaCl₃, affects the growth response, but no evidence is provided to show that this treatment blocked the calcium signal change. Ideally, additional calcium perturbations would also be tested to rule out nonspecific effects.

We believe these concerns have been addressed by the previously mentioned new data on hyponasty and GCaMP3 fluorescence for Ca²⁺ using LaCl₃ as well as a second Ca²⁺ blocker (Verapamil), further confirmed by the *glr3.1glr3.3glr3.6 X GCaMP3* cross.

Finally, one must show that driving a calcium flux (in the absence of the touch stimulus) is sufficient to cause the downstream response. Otherwise, one can only emphasize a correlation of the response with calcium flux.

Interesting and challenging suggestion. It is very difficult to induce a calcium flux without doing anything to the plant. Yet, we managed to apply a calcium agonist, Mastoparan, to the leaf tip, and compared this treatment to a Mock control. We found that Mastoparan indeed induces a cytosolic calcium increase and triggers a hyponastic response of approximately 12 degrees. Please see supplemental figure 6a and b. This is now also mentioned in the manuscript text lines 171-176.

Abstract: Based on data presented, it is too strong to say that perturbation of the calcium response and the absence of trichomes “inhibit” touch-induced hyponasty.

We replaced “inhibit” with “reduced”.

Methods: What was the rationale to choose ZT=2hr? Is there a role for the circadian clock?

Leaf angles do indeed change throughout the diurnal cycle, so it is important to standardize the methods to a specific time of day. ZT = 2 gives us sufficient time during the photoperiod to perform measurements and analyses. We are not aware of studies indicating a specific involvement of the circadian clock in touch-induced hyponasty, and it would be an interesting avenue for future studies.

REVIEWER COMMENTS

Reviewer #1 (Remarks to the Author):

This review is for the revised version of the manuscript entitled "Mechanodetection of neighbor plants elicits adaptive leaf movements through calcium dynamics". The authors further validated their conclusions according to the reviewers' comments and suggestions by demonstrating a substantial amount of new data. However, there are some concerns that need to be addressed.

1) The authors generated transgenic plants expressing UBQ10pro:GCaMP3 in the *glr3.1glr3.3glr3.6* and the *gl1* backgrounds. As GCaMP3 fluorescence is highly dependent on the expression and/or protein levels of GCaMP3, so the authors should perform quantitative PCR and western blotting using those plants.

2) Supplemental Figure 6a: Mock treatment clearly induced the GCaMP3 fluorescence in the leaf tip. The authors should show the mock graph to compare with mastoparan-treated samples.

3) The authors argue that mechanostimulation of trichomes in the leaf tip induces a hyponastic response, and the new genetic data support this hypothesis well. However, most of the hyponasty experiments were artificially performed using a transparent tag, by which the mechanical forces are likely applied not only to trichomes but also to a main vein or other places. Is the hyponastic response induced by stimulating only trichomes with toothpicks or brushes?

Reviewer #2 (Remarks to the Author):

The authors have significantly expanded and improved the manuscript. I only have one comment regarding the transcriptomics comparison with other previous studies. The authors state on line 111 'while there was no significant overlap with MYC2 MYC3 MYC4-dependent response (Supplementary Fig. 3c).'

I don't see where this conclusion comes from. The Fisher exact test p-value is 0.0005 in the legend of Fig S3C, which looks significant. The authors should rather look at the 266 *myc234*-dependent touch genes mentioned in the Van Moerkercke 2019 study, or the 82 core MYC2-regulon genes. A supplementary table with the genes in the overlaps from Figure S3 should be provided so that the readers can see which genes are actually in common between the datasets.

Also, I did not find the excel sheet file 1_DEGS that the rebuttal letter mentions in the files accessible to the reviewers, so I could not see the identity of the 35-15 etc genes, or of any of the DEGs. Please provide a full excel document with all gene expression and statistical analysis data from this study, and the comparisons with the other studies.

Reviewer #1 (Remarks to the Author):

This review is for the revised version of the manuscript entitled "Mechanodetection of neighbor plants elicits adaptive leaf movements through calcium dynamics". The authors further validated their conclusions according to the reviewers' comments and suggestions by demonstrating a substantial amount of new data. However, there are some concerns that need to be addressed.

1) The authors generated transgenic plants expressing UBQ10pro:GCaMP3 in the glr3.1glr3.3glr3.6 and the gl1 backgrounds. As GCaMP3 fluorescence is highly dependent on the expression and/or protein levels of GCaMP3, so the authors should perform quantitative PCR and western blotting using those plants.

First of all, we did not do new transfections but crossed the *UBQ10pro:GCaMP3* reporter to the mutant backgrounds. Therefore, the genomic insertion site is identical between the different backgrounds and expression of sensor, driven by the ubiquitous UBQ10 promoter should be expected to be constant as well.

Nevertheless, we did indeed perform the requested qPCR, and decided to perform an alternative analysis instead of the Western that should satisfy the comment. Both analyses confirm that sensor expression is indeed stable between the backgrounds (see Figure R1 below).

Firstly, we took the original fluorescence data and quantified the sensor baseline absolute fluorescence prior to mechanostimulation in the different backgrounds, which would represent the functional sensor protein levels. We found no differences between the backgrounds for the absolute fluorescence, indicating similar sensor protein levels (Figure R1A,B). Indeed, as suggested by the reviewer, we still also performed qPCR experiments and these confirmed that also at the gene expression level there is no difference between the backgrounds for expression of the sensor (Fig. R1B).

Fig. R1: GCaMP3 expression is similar between Col-0, *gl1* and *glr3* triple (*glr3.1 glr3.3 glr3.6*) mutant background. (A) Absolute fluorescence (F₀) signal of the GCaMP3 reporter in Col-0 versus *glr triple* and Col-0 versus *gl1* backgrounds. (B) Expression of GCaMP3 relative to *AT5G25760* (*UBIQUITIN-CONJUGATING ENZYME 21*) in a Col-0 (GCaMP3), *gl1* or *glr triple* background.

2) Supplemental Figure 6a: Mock treatment clearly induced the GCaMP3 fluorescence in the leaf tip. The authors should show the mock graph to compare with mastoparan-treated samples.

Good point, this has now been included. Indeed, mock treatment, because of physically puffing a droplet of liquid on the leaf tip, leads to a very transient response of the sensor, and it can be seen in the revised graph in supplemental figure 6A that mastoparan gave a faster and more persistent induction, consistent with its function.

3) The authors argue that mechanostimulation of trichomes in the leaf tip induces a hyponastic response, and the new genetic data support this hypothesis well. However, most of the hyponasty experiments were artificially performed using a transparent tag, by which the mechanical forces are likely applied not only to trichomes but also to a main vein or other places. Is the hyponastic response induced by stimulating only trichomes with toothpicks or brushes?

There are two responses to this question.

1. We performed the proposed experiment and indeed observe a very modest induction of hyponasty in response to repeated brushing with a brush. These data have now been added to supplemental figure 9F. Measurements were made just minutes before lights off in the growth chamber, after 7 hours of manual trichome stimulation because manual brushing at 10 minute intervals is impossible through the dark period, so a prolonged treatment was not possible.
2. If the interpretation from our text was that trichome mechanosensing is the only factor in touch-induced hyponasty, then we worded our conclusions too strongly. We show that trichomes are clearly very important, and the first tissue to touch a neighbor, but when a leaf touches a neighbor leaf or an inert object, there would also be a change of mechanical forces beyond the trichome that could certainly also contribute to the response. We have revised some of our wording in the manuscript (especially the subheading of this section) to leave room for such interpretations.

Reviewer #2 (Remarks to the Author):

The authors have significantly expanded and improved the manuscript. I only have one comment regarding the transcriptomics comparison with other previous studies. The authors state on line 111 'while there was no significant overlap with MYC2 MYC3 MYC4-dependent response (Supplementary Fig. 3c).'

I don't see where this conclusion comes from. The Fisher exact test p-value is 0.0005 in the legend of Fig S3C, which looks significant. The authors should rather look at the 266 myc234-dependent touch genes mentioned in the Van Moerkercke 2019 study, or the 82 core MYC2-regulon genes. A supplementary table with the genes in the overlaps from Figure S3 should be provided so that the readers can see which genes are actually in common between the datasets.

We have included the additional comparison with the MYC2 regulon, as suggested (Supplemental figure 3F). We also revised the description in the legend and made a small revision in the Results description to prevent any ambiguity. We have also included a supplemental excel file (excel file 2_comparisons.xlsx) with genes in the VENN diagram overlaps as per the reviewer's request. We appreciate this suggestion; this is very helpful for the readers.

Also, I did not find the excel sheet file 1_DEGS that the rebuttal letter mentions in the files accessible to the reviewers, so I could not see the identity of the 35-15 etc genes, or of any of the DEGs. Please provide a full excel document with all gene expression and statistical analysis data from this study, and the comparisons with the other studies.

Thank you for catching this. It looks like this file somehow failed to upload in the previous revision, for which we apologize. The file is included in this revision. The excel file with the comparisons with other studies, including the Fisher exact values is included as a separate excel file as mentioned in response to the previous point.